



# Multi-level emulation of a volcanic ash transport and dispersion model to quantify sensitivity to uncertain parameters

Natalie J. Harvey[1], Nathan Huntley[2], Helen Dacre[1], Michael Goldstein[2], David Thomson[3], and Helen Webster[3]

[1]Department of Meteorology, University of Reading, Reading, RG6 6BB, UK
[2]Department of Mathematical Sciences, Durham University, Durham DH1 3LE, UK
[3]Met Office, FitzRoy Road, Exeter, EX1 3PB, UK

*Correspondence to:* Natalie Harvey (n.j.harvey@reading.ac.uk)



**Abstract.** Following the disruption to European airspace caused by the eruption of Eyjafjallajökull in 2010 there has been a move towards producing quantitative predictions of volcanic ash concentration using volcanic ash transport and dispersion simulators. However, there is no formal framework for determining the uncertainties on these predictions and performing many simulations using these complex models is computationally expensive. In this paper a Bayes linear emulation approach is applied to the Numerical Atmospheric-dispersion Modelling Environment (NAME) to better understand the influence of source and internal model parameters on the simulator output. Emulation is a statistical method for predicting the output of a computer simulator at new parameter choices without actually running the simulator. A multi-level emulation approach is applied to combine information from many evaluations of a computationally fast version of NAME with relatively few evaluations of a slower, more accurate, version. This approach is effective when it is not possible to run the accurate simulator many times and when there is also little prior knowledge about the influence of parameters. The approach is applied to the mean ash column loading in 75 geographical regions on 14 May 2010. Through this analysis it has been found that the parameters that contribute the most to the output uncertainty are initial plume rise height, mass eruption rate, free tropospheric turbulence levels and precipitation threshold for wet deposition. This information can be used to inform future model development and observational campaigns and routine monitoring. The analysis presented here suggests the need for further observational and theoretical research into parameterisation of atmospheric turbulence. Furthermore it can also be used to inform the most important parameter perturbations for a small operational ensemble of simulations. The use of an emulator also identifies the input and internal parameters that do not contribute significantly to simulator uncertainty. Finally, the analysis highlights that the fast, less accurate, version of NAME can provide useful information without needing the accurate version at all. This approach can easily be extended to other case studies, simulators or hazards.

# 1   Introduction

Volcanic ash is a significant hazard to aircraft, and human life, by reducing visibility and causing both temporary engine failure and permanent engine damage (Casadevall, 1994). The presence of ash disrupts air traffic and can result in large financial losses to the aviation industry. The eruption of the Icelandic volcano



Eyjafjallajökull in April 2010 disrupted European airspace, the busiest airspace in the world, for thirteen days, grounded over 95,000 flights (European Commission, 2011) and is estimated to have cost the airline industry €3.3 billion (Mazzocchi et al., 2010).

In the event of an eruption, the decision to fly is informed by information provided by one of the nine Vol-5 canic Ash Advisory Centres (VAACs). The VAACs issue hazard maps of predicted ash cloud extents based on forecasts from Volcanic Ash Transport and Dispersion simulators (VATDs). After the large-scale disruption caused by the 2010 Eyjafjallajökull eruption new guidelines were brought in by EUROCONTROL (the European Organisation for the Safety of Air Navigation) which require predictions of ash concentration values as well as ash cloud extents. However, there are large uncertainties in the VATD ash concentration 10 forecasts. These uncertainties arise from a number of sources including incomplete or inaccurate knowledge of the specific volcanic eruption (source uncertainty) and meteorological conditions and other sources of parameter and forcing function uncertainty, as well as particular physical processes being simplified or omitted (structural uncertainty) in any particular simulator. Currently, no systematic estimation of the resulting uncertainty is performed. This is a major limitation of the operational system and as such there is the danger of 15 making incorrect decisions due to misjudging the accuracy of the simulator predictions.

There have been many studies investigating the processes that control the long-range dispersion of volcanic ash. The majority of these studies focus on a small number of simulator inputs or parameters and change the parameters one-at-a-time (OAT) to assess their impact on the predictions of volcanic ash transport. These studies test the difference between the simulator output from a control or baseline case and the output from 20 the perturbed cases. This approach is appealing as it always calculates the change in the simulator away from a well known baseline. Examples of studies that use this approach are Costa et al. (2006); Witham et al. (2007); Webley et al. (2009); Dacre et al. (2011); Devenish et al. (2012a, b); Folch et al. (2012); Grant et al. (2012); Witham et al. (2012b); Dacre et al. (2015). However, there are three main disadvantages of using OAT analysis. First, the amount of parameter space that is sampled quickly reduces as the number of pa-25 rameters considered is increased (Saltelli and Annoni, 2010). Secondly, OAT tests ignore any interactions between parameters. For example it is possible that perturbing two parameters separately in OAT tests might lead to negligible impacts, while the impact produced by perturbing them together might be much larger. Finally, the analysis cannot provide an overall assessment of uncertainty.





Performing sensitivity tests that cover a wide range of parameters and parameter values for a complex simulator, such as a VATD simulator, is expensive in both time and money. This makes uncertainty quantification impractical as one can only afford a limited amount of simulator runs. Uncertainty and sensitivity analyses as well as calibration require a large number of runs. In this study we introduce the use of emulation to

understand the sensitivity of an operational VATD simulator to source and internal simulator parameters.

An emulator is a simple statistical approximation of a complicated and (typically) computationally-expensive function, such as a computer simulator, that can be evaluated almost instantly over the whole parameter space. The emulator provides a prediction for the simulator's output at any given parameter choice, and an associated uncertainty for this prediction (this can take the form of a full probability distribution, or an ex-

pected value and variance). This enables the quantification of the impact of each simulator parameter on the prediction of the dispersion of volcanic ash. This approach has been used successfully in tsunami modelling (Sarri et al., 2012), simulating convective cloud (Johnson et al., 2015), aerosol modelling (Lee et al., 2011, 2012, 2013), galaxy formation (Vernon et al., 2010) and regional climate projections (Harris et al., 2010).

Emulators have several main uses in analysing computer simulators. They can be used for calibration, to

determine which parameters lead to simulator output that reasonably matches observed data. They can also be used for forecasting the future behaviour of the system in question. Finally, as in this paper, they can be used as a research tool to better understand the simulator, the role of the parameters, the interactions between them and to help guide future research priorities.

The aim of this paper is to demonstrate the potential of the emulation approach applied to a VATD simula-

tor. We use the Numerical Atmospheric-dispersion Modelling Environment (NAME) developed at the UK Met Office (Jones et al., 2007). This simulator is used as the operational model at the London VAAC and can predict the location and concentration of volcanic ash following a volcanic eruption. In this study we focus on predicting the vertically integrated (or column) mass loadings in a particular geographical region which occured following the 2010 Eyjafjallajökull eruption. The goal is to identify which parameters are the

principle drivers of the uncertainty in the simulator's predictions of column loadings, and to investigate how exactly these parameter values influence the output. The emulators used are also designed for use in history matching, which is a method for determining which parameters give plausible matches to observations. This application of the emulators is deferred to a future article.



The paper is structured as follows. Section 2 describes the NAME simulator and the case study. Section 3 details the parameters that are varied in this study and the plausible ranges (as assessed by the simulator experts) for these parameters. Section 4 describes the choice of simulator runs used to build the emulators, and the simulator outputs that are to be emulated. Section 5 gives an overview of the statistical methods used in the analysis. The application of these methods to the case study is detailed in Sec 6.

## 2 Description of NAME and chosen case study

### 2.1 Model description

NAME is the VATD simulator used by the London VAAC. It is a Lagrangian particle dispersion model originally developed in response to the 1986 Chernobyl disaster. Particles, each representing a mass of volcanic ash, are released from a source. These particles are advected by 3D wind fields provided by forecasts or analyses from a numerical weather prediction (NWP) model. The effect of turbulence is represented by stochastic additions to the particle trajectories based on estimated turbulence levels. NAME also includes parameterisations of sedimentation, dry deposition and wet deposition which are required to simulate the dispersion and removal of volcanic ash. The ash concentrations are calculated by summing the mass of particles in the model grid boxes and over a specified time period. It is important to note that some processes affecting the eruption plume are not represented in NAME or not included the NAME configurations used in this study. Missing processes include aggregation of ash particles, near source plume rise and processes driven by the eruption dynamics (e.g. Woodhouse et al., 2013). Note that the simulations presented in this paper were performed using NAME version 6.1.

To predict the transport and dispersion of ash, information about the volcanic eruption is required. These are known as eruption source parameters (ESPs) and include plume rise height, mass eruption rate, vertical profile of the plume emissions, particle density and particle size distribution. ESPs are required to initialise the NAME simulations. Full details of the NAME setup used by the London VAAC can be found in Witham et al. (2012a). The simulations used in this study have a start time of 2300 UTC on 7 May 2010. This start time has been chosen to ensure that NAME has had sufficient time to spin up before the chosen



case study. The details of the other model parameters is discussed in Sect. 3. The ash column loadings are calculated by summing the mass of the ash in model grid boxes and averaged over 1 hour. Here model grid

boxes are 0.375° latitude by 0.5625° longitude (approximately 40 km x 40 km).

## 2.2   Case study description

The case study chosen here is that of 14 May 2010. This is during the later phase of the Eyjafjallajökull eruption (14 April – 23 May). Although this later phase of the eruption did not have as much impact on the aviation industry, it is very well observed using ground-based instruments (e.g. Pappalardo, 2013), aircraft

measurements (e.g. Johnson et al., 2012) and satellites (e.g Francis et al., 2012). Due to the large amount of observational data it is also the focus of several modelling studies (e.g. Grant et al., 2012; Devenish et al., 2012a; Dacre et al., 2013). Between the 12 and 14 May, a low pressure system moved across Iceland transporting ash cyclonically to the North and West of Iceland on 12 May, towards Europe on 13 May and to the West of Iceland on 14 May. This followed a period (approximately 7 days) of relatively settled weather

dominated by a large area of high pressure in the the North Atlantic. The synoptic situation at 0000 UTC 14 May is shown in Figure 1a. Figure 1b shows a satellite infrared image taken by the AVHRR at 0613 UTC on the 14 May. There are high level clouds ahead of the occluded front located between Ireland and England. Behind the front there is low-level stratus cloud.

## 3   Choice of uncertain input parameters

Five eruption source parameters and nine internal simulator parameters were selected to represent the main uncertainties affecting the simulation of the dispersion of the volcanic ash in the NAME simulator. A short description of each parameter is given below along with an associated plausible range. The range represents our assessment of uncertainty on the value of each parameter. It is within these ranges that the training runs of the simulator will be performed in order to build the emulators. The uncertainty assessments were found through a small expert elicitation exercise in which information from relevant literature was combined with expert knowledge of NAME and its parametrisation schemes. Table 1 summarises the parameters and their



plausible ranges. In this study we do not consider the impact of the meteorological data used to drive NAME. More detailed expert judgements on the relative plausibility of parameter choices are not required to build

an emulator, although if available could be used to adjust the training design.

## 3.1 Eruption source parameters

This section describes in detail the parameters specific to the eruption source and how they are perturbed in the runs used to build the statistical emulator.

### 3.1.1 Plume height, H

Plume height governs the height at which the ash particles are emitted into the atmosphere. This can have a large impact on the horizontal and vertical structure of the ash cloud as atmospheric wind speed and direction vary with height. Therefore to simulate realistic dispersion following an eruption it is necessary to know this height as accurately as possible. During the 2010 Eyjafjallajökull eruption information about the plume height was available from the Iceland Meteorological Office's C-band radar based at Keflavík

Airport. However, there are time periods when no radar data was available. This was due to a variety of factors including the plume being obscured by meteorological cloud, missing radar scans and the fact that when the plume height was below 2.5 km it could not be detected due to the orography in the local area. When no observational plume height is available the last observed value persists until a new observation is made. In this study we will be using the data from the Keflavík radar (Arason et al., 2011) as the control

plume height. This control height is then perturbed by an increment in each of the simulations used to build the emulator. The maximum and minimum increment used is ±2 km. This is in line with observational error from the radar.

### 3.1.2 Vertical distribution of ash

In this study there are two possible vertical profiles of ash: full depth source and thin layer source (referred to in Grant et al. (2012) as "uniform" and "top hat"). In the full depth case all the ash is evenly distributed



from the volcano vent to the plume height. In the thin layer case all of the ash is emitted uniformly in a thin layer, with thickness $dz$. The middle of this thin layer is coincident with the maximum height of the plume. In this study 1700 full depth runs and 1700 thin layer runs have been performed. In the thin-layer cases $dz$ has been determined by perturbing a control value of $dz$ equal to 1 km. The minimum/maximum $dz$ possible is 0.1 km/2 km. This range spans the observed ash layer depths in the literature for our case study (e.g. Marenco et al., 2011; Schumann et al., 2011; Grant et al., 2012; Pappalardo, 2013; Dacre et al., 2015).

### 3.1.3 Mass eruption rate, MER

Currently there is no direct method of measuring how much mass is being emitted from an erupting volcano. Therefore many VAACs use an empirical relationship between the observed plume height and the eruption rate. There are number of relationships in the literature relating these two quantities (e.g Sparks et al., 1997; Mastin et al., 2009). In this paper the following relationship, based on Mastin et al. (2009) is used:

$$MER = 140.8H^{4.15}, \tag{1}$$

where $H$ is the plume height above the volcano summit in kilometers and MER represents the total mass eruption rate in kilograms per second (Mastin et al., 2009; Webster et al., 2012). Here $H$ is the perturbed plume height described in Sect. 3.1.2. Due to the empirical nature of this formulation the MER also has an associated uncertainty as the data used to form the relationship is based only on a small number of volcanoes of a similar nature (Mastin et al., 2009). To account for this the MER is perturbed by a factor between 1/3 and 3.

### 3.1.4 Particle size distribution, PSD

In the simulations used here, only fine ash is represented with diameters ranging from 0.1–100 $\mu$m separated into 6 size bins. The NAME default PSD (shown in Table 2) is based on observations by Hobbs et al. (1991) of ash from explosive eruptions of Mount Redoubt, St Augustine and Mount St Helens. The mass fraction of dispersing material is divided over the model particles within each size range. Each model particle may


correspond to many actual particles of a certain diameter. The exact diameter allocated to each model particle is such that the log of the diameter is uniformly distributed within each size range making up the PSD.

The PSDs used in the runs to build the emulator were formulated as follows. Dacre et al. (2013) present
several observed PSDs for the period around 14 May 2010; it was decided to choose a range for the PSDs that included all of these. These alternatives can all be reasonably reproduced using gamma distributions with particular shape and scale parameters. Therefore, instead of specifying a range for the frequency associated with each particle diameter bin, a range was specified for these two parameters. For any given pair within this range, the required PSDs can easily be computed. The range for these parameters was chosen such that
all the alternative PSDs could be reconstructed to a reasonable approximation.

### 3.1.5 Particle density

By default, the London VAAC modelling procedure assumes that ash particles are spherical and have a density of $2300 \ \mathrm{kg \ m^{-3}}$ (Bonadonna and Phillips, 2003). In this study the density is perturbed in the range $1350 \ \mathrm{kg \ m^{-3}}$ – $2700 \ \mathrm{kg \ m^{-3}}$. This range of perturbation to the particle density is considered to include the
15 uncertainty attributed to the particle shape and aggregation.

### 3.2 Internal simulator parameters

The long-range transport of volcanic ash can be described by two sets of processes. The first set, advection and dispersion, represent the motion of the particles. The second set, loss processes, model how the ash is removed from the atmosphere. This section describes in detail the parameterisations and associated parameters in NAME that represent the two sets of processes.

### 3.2.1 Advection and dispersion parameters

In NAME particles are advected in three dimensions by winds usually provided by a NWP model, with turbulent dispersion simulated by a random walk technique which represents the turbluent velocity structures





in the atmosphere. Particles are advected each time step with the change in position involving contributions
from the resolved wind velocity, the turbulence, and the unresolved mesoscale motions.

### 3.2.2   Free tropospheric turbulence

The diffusion due to free tropospheric turbulence is specified by a diffusivity, $K$, which is related to the turbu-
lent velocities and time scales of atmospheric motions. In NAME, the along-wind and cross-wind spread are
assumed to be equal, and the eddy diffusivity is further assumed to take the form $K = (\sigma_u^2\tau_u, \sigma_u^2\tau_u, \sigma_w^2\tau_w)$
where $\sigma_u$ and $\sigma_w$ are the standard deviations of the horizontal and vertical velocity fluctuations, respectively,
and $\tau_u$ and $\tau_w$ are the corresponding horizontal and vertical Lagrangian timescales. While these quantities
are likely to vary in space and time, NAME simply assumes fixed values. The default values and plausible
ranges for these parameters (see Table 1) are based on observations of vertical and velocity variances and dif-
fusivities above the atmospheric boundary layer and values used in other dispersion models (Schumann et al.,
1995; Dürbeck and Gerz, 1995, Webster and Thomson, personal communication). The upper limits of these
parameters are representing plausible extreme values of turbulence. Note that in this study the horizontal and
vertical free tropospheric turbulence parameters are varied by the same proportion.

### 3.2.3   Unresolved mesoscale motions

Low frequency horizontal eddies with scales that lie between the resolved motions of the input meteorolog-
ical data and the small three-dimensional turbulent motions represented in the turbulence parameterisation
scheme are parameterised separately by the unresolved mesoscale motion scheme (Webster et al., 2015). As
in the free tropospheric turbulence scheme the parameters governing the unresolved mesoscale motions are
fixed in time and space. It is assumed that the impact of the unresolved mesoscale motions is the same in
both components of the horizontal motion. The default values appropriate to the global NWP data used in
this study are $\sigma_m = 0.8$ m s$^{-1}$ and $\tau_m = 14400$ s. These default parameters are derived from the spectral
characteristics of the input meteorological data (Webster and Thomson, 2005). At long range, only the dif-
fusivity $\sigma_m^2\tau_m$ matters and so, to simplify the experimental design, we seek to perturb this without worrying



about the values of $\sigma_m$ and $\tau_m$ separately. To achieve this diffusivity range, between 0.05 and 2 times the default value, we kept $\tau_m$ constant at 6120 s and varied $\sigma_m$ from $0.27 - 1.74 \, \mathrm{m \, s^{-1}}$ as in Table 1.

### 3.2.4 Loss process parameters

This section describes the parameters associated with the processes that remove ash from the atmosphere. The loss processes represented in NAME are wet deposition and dry deposition (including sedimentation). Within NAME these losses are applied on a particle basis (i.e. the mass of each particle is reduced each time step).

### 3.2.5 Wet deposition

Wet deposition is the process of ash depletion by precipitation in the atmosphere. Two main processes are involved: washout, where material is "swept out" by falling precipitation, and rainout, where ash is absorbed directly into cloud droplets as they form by acting as cloud condensation nuclei. Both of these processes are parameterised in NAME using a bulk parameterisation. The removal of ash from the atmosphere by wet deposition processes is based on the depletion equation

$$\frac{dC}{dt} = -\Lambda C, \tag{2}$$

where $C$ is the ash concentration, $t$ is time and $\Lambda$ is a scavenging coefficient. The scavenging coefficient, $\Lambda$, is given by

$$\Lambda = Ar^B, \tag{3}$$

where $r$ is the precipitation rate in $\mathrm{mm \, hr^{-1}}$ and A and B are parameters which vary for different types of precipitation (e.g. rain or snow) and which process is being represented (e.g. washout or rainout). The values for A and B are based on observations and detailed cloud modelling (Maryon et al., 1999). Note that a review of the literature highlighted that the range of experimental values for snow is much more uncertain than for rain. This translates into a larger range of possible values of A and B for snow than rain.




In NAME the wet deposition scheme is only used if the prepitation rate is greater than a threshold value,
ppt_crit. This acts as a filter to light drizzle. The reason for applying this threshold is that historically
there has been an excessive light drizzle issue in the global version of the UK Met Office NWP model
(Webster and Thomson, 2014). Applying this threshold ensures that there is not an artifical over prediction
of wet deposition. The default value for ppt_crit is $0.03$ mm hr$^{-1}$. In this study this threshold is varied
between 0 and $0.1$ mm hr$^{-1}$.

### 3.2.6 Dry deposition

Dry deposition is the process by which material is removed from the atmosphere by transport to, and sub-
sequent uptake by, the ground in the absence of precipitation. Dry deposition in NAME is parameterised
through a deposition velocity, $v_d$. The flux of ash to the ground, $F$ is proportional to the near-surface con-
centration of ash, $C$, and is given by

$$F = v_d C \qquad (4)$$

where $v_d$ is determined using a resistance analogy.

$$v_d = \frac{1}{R_a + R_b + R_c}, \qquad (5)$$

where $R_a$ is the aerodynamic resistance, $R_b$ is the laminar sublayer resistance and $R_c$ is the surface resistance
(taken to be zero for particulates such as ash) (Webster and Thomson, 2011). The aerodynamic resistance,
$R_a$, is used to specify the efficiency with which the ash is transported to the ground by turbulence. It is
parameterised using a flux gradient approach and similarity theory (Maryon et al., 1999). This means that
the parameterisation is strongly influenced by the prevailing meteorological conditions, and thus $R_a$ is per-
turbed using a scaling factor between 0 and 2. The laminar sublayer resistance, $R_b$, represents the resistance
to transport across the thin quasi-laminar layer adjacent to the surface. It is determined by both the meteo-
rological situation and particle size. The parameterisation follows the work of Underwood (2011). For small
particles, smaller than 1 $\mu$m,

$$R_b = \frac{300}{u_*}, \qquad (6)$$



where $u_*$ is the friction velocity and for larger particles

$$R_b = \frac{300}{u_* D^2},$$  (7)

where $D$ is the particle diameter in $\mu$m. In this study the numerator of Eq 6 and Eq 7 is varied between 0 - 300 to represent the range of uncertainty in the value of $R_b$.

### 3.2.7  Sedimentation

Sedimentation of ash is represented in NAME using a sedimentation velocity, $w_{sed}$. This velocity is calculated using the particle diameter ($D$), particle density ($\rho_p$) and ambient meteorological variables at the particle location (see Maryon, 1997; Webster and Thomson, 2011). In this study, $w_{sed}$ is not perturbed as it is assumed that changes in PSD and particle density cover the range of plausible sedimentation velocities.

### 3.2.8  Distal fine ash fraction

The true particle size distribution of ash particles emitted during an eruption includes extremely large particles that fall to the ground very quickly. For forecasting the effects of the eruption on aviation only the particles that will be transported large distances need to be considered. These particles form the distal ash cloud. The fraction of the total emitted ash that remains in this cloud is defined as the distal fine ash fraction (DFAF). DFAF is difficult to determine as it requires accurate measurements of the particle size distribution
and understanding of any aggregation processes occurring. It is also possible for DFAF to vary over time and in different parts of the ash cloud. Estimates of DFAF for the 2010 Eyjafjallajökull eruption range from 0.7 – 18.5% (Dacre et al., 2011; Grant et al., 2012; Devenish et al., 2012b; Dacre et al., 2013). The default DFAF assumed by the London VAAC is 5% (Witham et al., 2012b). DFAF simply scales the modelled ash concentration and therefore does not need to be included in the analysis in this paper as the impact on the simulator output is understood perfectly.




## 4 Simulator runs and simulator outputs

In this study attention is focused on the ash cloud on 14 May 2010. The simulator has been set up to provide ash predictions every hour at a resolution of 0.375° latitude by 0.5625° longitude (approximately 40 km x 40 km). Fig. 2(a) shows the simulated ash column loading at 0000 UTC on 14 May 2010 for a choice of parameters near the default values. High column loadings are found near, and to south east of the volcano. The main plume extends towards the United Kingdom with an area of relatively low column loading in the Atlantic west of Ireland. Rather than attempt to model the entire ash cloud, it was decided to restrict attention to a small number of summaries, specifically the average ash column loading predicted over 75 large areas (up to four regions per hour for a total of 24 hours). These areas were chosen to cover the geographical regions where large amounts of ash were detected by satellite observations on this day. The ash column loadings retrieved using SEVIRI satellite data at 0000 UTC on 14 May 2010 are shown in Fig. 2(b). The regions used for the first hour are marked by the black boxes. The list of all regions used in the calculations can be found in Table 3.

NAME is not a fast simulator (each run of the simulator for this study took between half an hour and an hour), so it is not possible to evaluate it for very many different parameter sets. The number of NAME runs that were feasible was potentially insufficient to build the statistical models of interest. However, a fast approximation of the standard NAME output could be generated by reducing the number of particles used in the simulator from 10,000 per hour to 1,000 per hour. We expect the effect of this 10-fold reduction in particle numbers to increase the partcile-sampling noise in the simulations by a factor of $\sqrt{10}$. This can provide many approximate runs to complement the relatively few standard simulator runs. Henceforth, the fast approximation is referred to as "the fast simulator" and the standard version is referred to as "the slow simulator".

1500 parameter sets were chosen for the fast simulator runs, using a maximin Latin hypercube design (Urban and Fricker, 2010). A Latin hypercube design is a method of generating multidimensional parameter sets, designed to ensure good coverage of the overall parameter space. For generating a sample size 1500 from a hypercube, the range of each individual parameter is divided into 1500 segments. Then, a random sample of size 1500 is generated such that, for each parameter, each of its 1500 segments includes





exactly one simulated value. A maximin Latin hypercube attempts to build such a hypercube with the largest minimum distance between any two generated parameter sets. This method is used rather than simply generating 1500 random parameter sets independently to ensure that the chosen points are more evenly spaced throughout the parameter space.

200 different parameter sets were chosen for the slow simulator runs in the same way. Finally, the fast simulator was also run at the same 200 points as the slow simulator, so the difference between the two simulators could be assessed. For some regions, there was an almost complete agreement between the two simulators, whereas for other regions, the two were related but not in agreement. Examples of these different relationships can be seen in Fig. 3. In all regions there was strong correlation between the two simulators, with many correlations being $0.99$, and none lower than $0.7$.

Before proceeding, some notation should be introduced. A particular parameter set is denoted by $\mathbf{x}$, and the $i$th parameter within this set is $x_i$. Collections of parameter sets are denoted by $\mathbf{x}_1, \ldots, \mathbf{x}_n$. The 200 parameter sets at which the slow simulator is evaluated are denoted by $\mathbf{x}_1, \ldots, \mathbf{x}_{200}$. and the remaining 1500 parameter sets are denoted by $\mathbf{x}_{201}, \ldots, \mathbf{x}_{1700}$. The sets of parameter sets are labelled

$$\mathcal{X}_S = \{\mathbf{x}_1, \ldots, \mathbf{x}_{200}\}$$
$$\mathcal{X}_F = \{\mathbf{x}_{201}, \ldots \mathbf{x}_{1700}\}.$$

Finally, each parameter set $\mathbf{x}$ is normalised so that each individual parameter value lies between 0 and 1.

The slow simulator is denoted by $f$ and the fast simulator by $f'$. $f(\mathbf{x})$ and $f'(\mathbf{x})$ can be seen as vectors of length 75 (the total number of geographical regions) with $f_i(\mathbf{x})$ being the value of the average ash column load in the $i$th region (for example, region 6 is the third region at 0100 UTC 14 May 2010—see Table 3). If $\mathcal{X}$ is a set of parameter sets, then $f(\mathcal{X})$ is the set of simulator outputs generated by applying $f$ to each element of $\mathcal{X}$. The set of simulated outputs $f(\mathcal{X}_S)$ (that is, the set of all slow simulator output) is denoted by $D$, and $f'(\mathcal{X}_S \cup \mathcal{X}_F)$ (the set of all fast simulator output) is denoted by $D'$.

In this notation, the goal is then to use the evaluations $D$ and $D'$ to make inferences about the value of $f(\mathbf{x})$ for any other parameter set $\mathbf{x}$. This will involve building a statistical approximation for $f$, termed an



*emulator*. The next section describes the general form of such a model, and the statistical framework used to
make inferences from the simulator outputs $D$ and $D'$.

## 5   Statistical methods

### 5.1   Emulation

An emulator is a simple statistical approximation of an expensive function $f(\mathbf{x})$, built using a (often small)
collection of simulator runs $f(\mathbf{x}_i)$, which can be thought of as "data" or "observations". There are several
desirable properties of an emulator:

- It must evaluate quickly.

- It must be expressive enough to provide good approximations to the simulator and to allow meaningful
  prior judgements.

- It should predict that $f(\mathbf{x})$ and $f(\mathbf{x}')$ should be very close when $\mathbf{x}$ and $\mathbf{x}'$ are very close.

A typical choice to satisfy these requirements for a scalar-valued $f(\mathbf{x})$ is

$$f(\mathbf{x}) = \sum \beta_i g_i(\mathbf{x}) + u(\mathbf{x}), \tag{8}$$

or for a vector-valued $f(\mathbf{x})$

$$f_i(\mathbf{x}) = \sum_j \beta_{ij} g_{ij}(\mathbf{x}) + u_i(\mathbf{x}).$$

For the rest of this section, attention is restricted to scalar-valued $f$ for simplicity of notation.

5  Here, $g_i(\mathbf{x})$ are known simple functions (for instance polynomials), and the $\beta_i$ are unknown coefficients.
These terms control the global trend of the model. The function $u(\mathbf{x})$ controls the local variation of the
model. Typically, it is supposed that $\mathrm{E}(u(\mathbf{x})) = 0$ and that $\mathrm{Corr}(u(\mathbf{x}), u(\mathbf{x}'))$ is some function of the distance





between $\mathbf{x}$ and $\mathbf{x}'$, such that the correlation falls as parameters get further apart. For example, a popular choices and the one used for this application is

$$\mathrm{Corr}\left(u(\mathbf{x}_1), u(\mathbf{x}_2)\right) = \exp\left(-\left(\frac{d(\mathbf{x}_1, \mathbf{x}_2)}{\delta}\right)^2\right),$$

where $d(\mathbf{x}_1, \mathbf{x}_2)$ is the Euclidean distance between the parameters, and $\delta$ is the *correlation length*, a parameter that determines how quickly correlation falls with distance. Finally, it is commonly assumed that $\mathrm{Var}\left(u(\mathbf{x})\right) = \sigma^2$ for all $\mathbf{x}$, so the variance of the local term is constant across the parameter space. Conceptually, the expectation, variance, and correlation are *a priori* uncertainty judgements.

15 Building an emulator therefore involves using a collection of simulator runs $f(\mathbf{x}_1), \ldots, f(\mathbf{x}_n)$ to

- identify the basis functions $g_i$;

- estimate the $\beta_i$;

- fit the residual function $u(\mathbf{x})$.

Such an emulator then provides predictions for $f(\mathbf{x})$ at a new $\mathbf{x}$. Since it is a statistical model, this prediction
20 also comes with an associated uncertainty, which will be low near observed simulator runs and higher away from them. Fig. 4 shows an emulator for a scalar-valued function of one variable.

There are many approaches to fitting such a model. Computer simulator applications often involve a mixture of observed simulator runs and expert knowledge, making a Bayesian framework a natural choice. However, specification of a full joint probability distribution for the problem is difficult and often leads to computational challenges. In the next section, some of these problems are summarised, and an alternative approach, *Bayes linear*, is described.

## 5.2 Bayes linear methods

Statistical analysis of computer simulators involves combining observations (for instance, simulator output
and real-world observations) and expert judgements (for instance, accuracy of simulator and accuracy of





observations). Such a problem naturally lies within the scope of Bayesian statistics, a popular and powerful tool for combining data with expert judgements, using Bayes theorem. A brief summary of the necessary components of such an analysis are as follows. From expert judgements, a joint probability distribution is constructed for $f(\mathbf{x})$ over the parameter space (for instance, through a joint probability distribution for the

$\beta$ and $u$ in Eq. (8)). A collection of simulator runs $f(\mathbf{x}_1)$, ..., $f(\mathbf{x}_n)$ would be made, and Bayes theorem used to calculate a posterior distribution for $f(\mathbf{x})$ at all values of $\mathbf{x}$. For calibration and forecasting, this would then be combined with a probability distribution across the parameter space (representing the relative plausibility of each $\mathbf{x}$ to experts), a distribution for the observation error, and a distribution representing the likely discrepancy between simulator output and reality—this step is outside the scope of this paper but will

be examined in a later article.

Such an approach has been successful in many applications. In complicated problems in high dimension, however, it has some drawbacks. A full Bayes calculation is computationally demanding and in high dimensions can be very sensitive to the initial prior specifications. Further, specifying the full high-dimensional probability distributions that properly reflect expert judgements is an extremely difficult task. Worse, the

complexity of the calculations makes it very hard to perform careful analysis to the sensitivity of the conclusions to these prior judgements. Often, these calculations will necessarily make use of computationally-convenient prior forms that do not correspond well with expert beliefs. Thus, the analysis will be sensitive to prior distributions that do not properly reflect our judgements, and the scale and nature of this sensitivity will be mostly unknown.

In this paper, the alternative *Bayes linear* approach is used (Goldstein and Wooff, 2007). As with a full Bayes analysis, the method combines prior judgements with observations through simple equations. Bayes linear analysis does not, however, require a full joint prior probability distribution specified for all variables. Rather, the experts need only to specify expectations, variances, and covariances for a few relevant quantities. Similarly, rather than a joint posterior probability distribution, Bayes linear analysis leads to adjusted

expectations, variances, and covariances for relevant quantities. Given a vector of data $D$ (for example, simulator runs $f(\mathbf{x}_1), \ldots, f(\mathbf{x}_n)$ that have been evaluated), the representation of $f$ in Eq. (8), and a vector of quantities of interest $B$ (for example, the value of the simulator $f(\mathbf{x})$ at some new $\mathbf{x}$ at which the simulator




has not yet been evaluated), the adjusted expectation and variance for $B$ are given by

$$\mathrm{E}_D(B) = \mathrm{E}(B) + \mathrm{Cov}(B,D)\,\mathrm{Var}(D)^{-1}(D - \mathrm{E}(D)) \tag{9}$$

$$\mathrm{Var}_D(B) = \mathrm{Var}(B) - \mathrm{Cov}(B,D)\,\mathrm{Var}(D)^{-1}\mathrm{Cov}(D,B). \tag{10}$$

Note that these equations hold for arbitrary $D$, not just the $D$ defined in the previous section (the set of slow simulator outputs). In particular, we will often replaced $D$ with $D'$ (the set of fast simulator outputs) in these equations.

By so reducing the complexity of the required prior judgements, it is easier to accurately represent these beliefs while retaining computational feasibility. Further, the relative simplicity of the adjustment process allows more convenient analysis of sensitivity to these judgements. On the other hand, the inferences from a Bayes linear analysis are not as expressive (expectations and variances, rather than a full probability distribution). Thus, a Bayes linear analysis is not simply an upgrade on a Bayesian analysis, but rather an alternative whose benefits and shortcomings must be carefully considered before deciding which to use, or whether to use a combination of the two.

The application of Bayes linear methods to an emulator requires prior judgements of expectations, variances, and covariances of the components of Eq. (8), that is, the quantities $\beta_i$ and $u(\mathbf{x})$. It is common to choose $\mathrm{E}(u(\mathbf{x})) = 0$ and $\mathrm{Var}(u(\mathbf{x})) = \sigma^2$ for all $\mathbf{x}$, and $\mathrm{Cov}(\beta_i, u(\mathbf{x})) = 0$ for all $i$ and $\mathbf{x}$. Thus, the total required specifications are

 – Expectation and variance matrix for $\beta$;

 – Correlation function $\mathrm{Corr}(u(\mathbf{x}_1), u(\mathbf{x}_2))$;

 – A value (or prior judgements) for $\sigma^2$.

These components are sufficient to apply Eqs. (9) and (10), with $B$ being $f(\mathbf{x})$ at some new $\mathbf{x}$, and $D$ being the observed simulator runs. Examples of this approach being successfully applied to computer simulators can be found in Craig et al. (1997); Vernon et al. (2010); Cumming and Goldstein (2009).

Even in this simplified form, it is often difficult to provide expert judgements about these quantities. With sufficient simulator runs, such as in the case of the fast simulator $f'$, the weight of the "observations" (that




is, the simulator output that has been seen so far) will be much greater than the prior judgements, so the adjusted expected values $E_D(\beta)$ will be driven primarily by the data, and their variances will be low. In such

a case, a successful method has been to use a standard (non-Bayesian) least-squares regression to estimate the $\beta$, and use the residual variance from the regression for $\sigma^2$. These results should be very similar to a Bayesian analysis, without needing to worry about the prior judgements for $\beta$.

The 200 runs of the slow simulator is on the borderline for such a method to work. The 1700 runs of the fast simulator should be enough to apply this simplification to an emulator for the fast simulator. Of course, the

fast simulator is not the simulator of true interest. However, it is likely that the fast simulator can provide useful information about the slow simulator. Hence, a method proposed in Cumming and Goldstein, 2009 is applied, in which the fast and slow simulators are linked through a simple model.

## 5.3    Linking fast and slow simulators

Recall that the fast simulator is $f'(\mathbf{x})$ and the slow simulator is $f(\mathbf{x})$. An emulator can be built for the fast

simulator:

$$f'(\mathbf{x}) = \sum_{i=1}^{p} \beta_i' g_i(\mathbf{x}) + u'(\mathbf{x}), \tag{11}$$

as follows. The $g_i$ are chosen by the analyst through exploration. The $\beta_i'$ are fixed by a least squares method, for instance the R function lm, to their least squares estimates $\hat{\beta}_i'$. $\mathrm{Var}(u'(\mathbf{x}))$ is taken to be the same for all $\mathbf{x}$ and is given by the residual variance from this least squares fit. The final component, the correlation $\mathrm{Corr}(u'(\mathbf{x}_1), u'(\mathbf{x}_2))$, can be fit using various methods; more details of this can be found in Appendix A1.1.

The next step is to link this to an emulator for $f(\mathbf{x})$ from Eq. (8). Notice that in Eqs. (8) and (11), the basis functions $g_i(x)$ are the same in both emulators. That is, it is supposed that the mean trend of the fast

simulator $f'$ has the same form (but different coefficients) as the simulator of interest $f$. If $f'$ is a reasonable approximation for $f$ (for instance, an older version of $f$ or a version of $f$ run at lower resolution) this supposition will usually be valid.

Further, the coefficients $\beta_i$ and $\beta_i'$ will often be similar. A model linking these coefficients will allow the fast simulator runs to provide information about the $\beta_i$. At the same time, this model must be flexible enough





that it does not impose a strong link where none exists. The same can be said of the link between $u(\mathbf{x})$ and
$u'(\mathbf{x})$. A simple model is

$$\beta_i = \rho_i \beta_i' + c_i$$
$$u(\mathbf{x}) = \rho_0 u'(\mathbf{x}) + w(\mathbf{x}),$$

where $\rho_0$, $\rho_i$ are unknown multipliers and $c_i$ are unknown scalars. If the two simulators are very similar,
then most $\rho_i$ will be near 1 and most $c_i$ will be near 0. If the value of $g_i(\mathbf{x})$ has a much smaller effect on
the fast emulator that on the slow emulator, $\rho_i$ will be much larger than 1. Where the value of $g_i(\mathbf{x})$ has
a much large effect on the fast emulator that on the slow emulator, $\rho_i$ will be near zero. If $g_i(\mathbf{x})$ has an
opposite effect on the fast emulator and the slow emulator, then $\rho_i$ will be negative. The emulation process
therefore involves using the fast simulator to work out the form of the emulator, to estimate the $\beta_i'$, and make
inferences about $u'$, and then using the slow simulator to make inferences about the $\rho_i$ and $w$. Note that
underlying this approach is the assumption that the slow simulator runs do not provide any more information
about the fast simulator.

In this application, it turned out that this could be further simplified to

$$\beta_i = \rho_i \beta_i'$$
$$u(\mathbf{x}) = \rho_0 u'(\mathbf{x}) + w(\mathbf{x}) \tag{12}$$

without noticeably reducing the effectiveness of the emulators.

This model requires prior expectations, variances, and covariances for the $\rho_i$ and $\rho_0$, as well as for $w(\mathbf{x})$. In
Appendix A1.2, more details of these prior requirements are provided.

With such a model and the relevant judgements, including the assumption that the $\beta_i'$ can be taken to be
the least squares estimates $\hat{\beta}_i'$, the Bayes linear adjustment for a new $f(\mathbf{x})$ can now be performed. This
calculation and the resulting equations are somewhat technical, so are given in Appendix A1.3; in particular
the adjusted expectation and variance can be found in Eqs. (A2) and (A3).





## 5.4 Diagnostics and validation

It is important to check that an emulator is performing well before using it to make predictions. There are several possible reasons an emulator would be poor. The form of the mean function could be missing an important term or even be totally misguided. The form of the correlation function might be inappropriate. The parameters in the correlation function (in this application, the correlation length) could be set at inappropriate values. Finally, some other assumptions, such as the assumption that $\text{Var}(u(\mathbf{x}))$ is the same for all $\mathbf{x}$, could be seriously misleading.

The mean function plays a large role in these emulators. The usual diagnostics from linear models can be valuable in assessing the adequacy of the chosen mean function. The $R^2$, a statistic that represents the proportion of variation explained by the parameters in the linear model, is a useful number to check first. If this is low, then the mean function is not explaining much of the variation in the simulator output, and adding new terms or changing the form of the mean function entirely should be considered. Examining the residuals can also be useful in this process, in particular whether there are regions of the parameter space where the residuals are systematically large in one direction.

A simple and effective method of validation is leave-one-out validation. In this procedure, all but one of the observed simulator runs are used to build an emulator, and this emulator is used to predict the one run that was left out. For $n$ simulator runs, this gives $n$ emulators and predictions. If the emulators frequently predict the left-out values to be far from the observed simulator run, this suggests a problem with the emulator. Here, "far from" means relative to the variance of the emulator—a useful rule of thumb is that about 95% of the validation runs should be within three standard deviations of the prediction.

If this proportion of successful prediction is far from 95%, this might signal a fundamental problem with the mean function and/or the form of the correlation function, but it can often simply signal a poor choice of correlation length. If the correlation length is too high, then the emulator variance will be too low and hence many observations will be judged "too far" from the emulator predictions. On the other hand, if the correlation length is too low, then the emulator will not be able to capture many patterns of local variation from the mean function that may be present (specifically, any such patterns that exist over distances much



higher than the correlation length). It is often possible to tune the correlation length so that the proportion of successful validations is around 95%.

## 6 Application to NAME

Throughout this section, unless otherwise specified, the quantity $f_1$, the average ash in the first region for the first hour, is being considered, and the full depth version of NAME is being used. The analysis was also run for the thin layer simulator, with similar results (although the emulators are slightly worse for this case).

### 6.1 Choosing basis functions and eliminating inactive parameters

The first stage of building an emulator is to choose the functions $g_i(\mathbf{x})$ in the mean trend. From experience, polynomial terms are often suitable choices. For each of the 75 outputs, linear models were built with i) first-order (linear) terms only; ii) second-order (quadratic) and first-order terms, with interactions; iii) third-order (cubic) and lower-order terms, with first-order interactions. Explicitly, these are the models

$$f'(\mathbf{x}) = \sum_i a_i x_i + u'(\mathbf{x})$$

$$f'(\mathbf{x}) = \sum_i a_i x_i^2 + \sum_i \sum_{j \neq i} b_{ij} x_i x_j + \sum_i c_i x_i + u'(\mathbf{x})$$

$$f'(\mathbf{x}) = \sum_i a_i x_i^3 + \sum_i b_i x_i^2 + \sum_i \sum_{j \neq i} c_{ij} x_i x_j + \sum_i d_i x_i + u'(\mathbf{x}),$$

where the $a_i$, $b_i$, $c_i$, $d_i$ collectively form the $\beta_i'$ in Eq. (11) (and, are of course, different values in the three different models). Note that "linear" in "linear model" refers to the linearity of the form $\sum_i \beta_i g_i(\mathbf{x})$, not the linearity of the $g_i$, so all three models here are linear models.

The adjusted $R^2$ was examined for each model. The findings of this procedure, when applied to the fast simulator runs for the full depth simulator, can be summarised as follows.





– The models with only first-order terms were inadequate in many cases, leading to low $R^2$ and high residual variance. For some of the outputs they did provide good fits (adjusted $R^2$ between 0.9 and 0.95).

– The second-order models were very good ($R^2$ over 0.95) for almost every region, and good for all regions (with the lowest $R^2$ of 0.89).

– The third-order models provide no noticeable improvements over second-order models.

As a result of this, the chosen $g_i$ were second-order and lower terms for all outputs.

The second stage of emulation is the removal of inactive parameters. In the linear model for any given output quantity, most of the parameters have little impact. Emulators can be improved by focusing on a few important parameters and leaving the rest out of the mean trend entirely. This involves adding a small "nugget" of variance into the emulator, uncorrelated with everything else. This nugget represents the fact that now the emulator does not exactly predict the simulator output even at parameters already sampled, because some parameters have been ignored. For example, if only parameters $x_1$ and $x_2$ are active, then the emulator will give the same prediction whatever the value of $x_3, \ldots$, whereas of course the simulator will give slightly different output in each case. The nugget accounts for this uncertainty. An estimate for the size of the nugget was derived by running the simulator with different values of the inactive parameters and observing the impact. This is a rather crude approach, but since the observed variation was several orders of magnitude lower than the other variances in the emulator, there is little benefit to a more careful analysis. Formally, the emulator becomes

$$f(\mathbf{x}) = \sum \beta_i g_i(\mathbf{x}_A) + u(\mathbf{x}_A) + v(\mathbf{x}),$$

where $\mathbf{x}_A$ are the active parameters, and $v(\mathbf{x})$ represents the nugget, with expectation zero, low variance, and zero correlation with everything else.

A policy of stepwise elimination was followed for each output: at each step, each parameter was removed in turn, and the change in $R^2$ was calculated. The parameter whose removal caused the smallest change in this was removed. This process was continued for each output until either 4 parameters were left or the removal of a single parameter would reduce the $R^2$ by more than 0.03. A third criterion, that the $R^2$



should not be allowed to fall below some critical value, was considered but turned out to be unnecessary. For most output quantities, this led to the emulators with four active variables, with more in a few of the 75 output areas. Parameters $x_1$ (plume height) and $x_3$ (mass eruption rate) were active in all models, with $x_7$ (standard deviation of free tropospheric turbulence) and $x_{12}$ (precipitation rate required for wet deposition)

active in most. Parameters $x_6$ (ash density), $x_{13}$ (scavenging coefficient parameter $A$ for rain), and $x_{15}$–$x_{18}$ (scavenging coefficient B and dry deposition resistances) were active in no emulators.

In a standard emulation this would conclude the removal of inactive parameters, but since in this case the fast emulator is to be linked to the slow emulator, it is important to check that there are no parameters being removed that are much more important for the slow emulator. For this reason, the same stepwise selection

was performed using the 200 runs of the slow simulator (ignoring the link with the fast emulator). This procedure selected the same parameters in most cases, occasionally with one difference. It is likely this is caused by small quasi-random differences in the $R^2$, but for safety these parameters were also added back into the emulators. This led to an extra parameter being activated for four of the outputs.

Finally, since parameters $x_4$ and $x_5$ were closely related (the parameters governing the gamma distribution

from which the particle size distribution was calculated), it was decided that an active $x_5$ should lead to an active $x_4$ as well. A summary of the number of times each parameter was active is shown in Table 4.

## 6.2   Emulating the fast simulator

Each of these linear models now gives an estimate for $\beta_i'$ and a residual variance that can be used for $\mathrm{Var}\left(u'(\mathbf{x})\right)$. Since 1700 is a large number of runs, it is reasonable to make the simplification that these

quantities are now known values. The only remaining task for the fast simulator's emulator is to specify the correlation. A squared correlation is used, that is,

$$\mathrm{Corr}\left(u'(\mathbf{x}_1), u'(\mathbf{x}_2)\right) = \exp\left(-\left(\frac{d(\mathbf{x}_1, \mathbf{x}_2)}{\delta'}\right)^2\right),$$

where $\delta'$, the correlation length, is to be set, and $d(\mathbf{x}_1, \mathbf{x}_2)$ is the distance between $\mathbf{x}_1$ and $\mathbf{x}_2$. In Appendix A1.1, some possibilities for choosing $\delta'$ are provided. Note that using a different scaling parameter

for each dimension of the parameter space can be necessary in many cases, but for this application a single



value proved sufficient (recall that all parameters have been normalised so they are all in $[0,1]$, otherwise different $\delta'$ would be needed for each dimension). The approach used in this application is to begin with $\delta' = 1/3$, then use leave-one-out validation using $f'(\mathcal{X}_F)$ to tune $\delta'$, and finally predict $f'(\mathcal{X}_S)$ using $f'(\mathcal{X}_F)$ and this $\delta'$ to check that the method has been successful.

This strategy suggested rather small values for the correlation lengths, between $0.1$ and $0.15$. Predictions of the remaining $200$ runs using the emulator built from the first $1500$ were accurate for all the outputs: an example can be seen in Fig. 5, for the case of the first output in the first hour. The emulator predictions are close to the observed output (that is, $f'(\mathcal{X}_S)$) relative to the emulator variances in most cases, and the emulator variances are small relative to the overall variability of simulator output across the parameter space.

This suggests that the emulator is a useful tool for prediction. The proportion of $f'(\mathcal{X}_S)$ predicted reliably (that is, within three standard deviations of the emulator variance) for each output ranged from 94.5% to 99%.

This analysis suggests that the emulator and the choices of $\delta'$ are appropriate. A final fast emulator was then built using all the runs ($f'(\mathcal{X}_S \cup \mathcal{X}_F)$) and the values of $\delta'$ calculated by the above method. The next step

is to link the fast simulator to the slow simulator and use the runs $f(\mathcal{X}_S)$ to make predictions for the slow simulator.

## 6.3  Emulating the slow simulator

The emulator for the fast simulator is linked to that of the slow simulator through Eqs. (12) (recall that the emulators for the slow and fast simulators are given by Eqs. (8) and (11) respectively). This requires prior judgements for $\rho_i$ and $w(\mathbf{x})$. For the latter, the judgements used were that $\mathrm{E}\left(w(\mathbf{x})\right) = 0$, $\mathrm{Var}\left(w(\mathbf{x})\right) =$

$\mathrm{Var}\left(u'(\mathbf{x})\right)$, and the correlation structure is the same as that of $u'(\mathbf{x})$, with correlation length $\delta$ initially set to $\delta'$ but explored later in the same way. Expectations, variances and covariances for $\rho_i$ were specified using the least-squares method in Appendix A1.2.

With these priors, and for a given value of $\delta$, the adjusted expectation and variance $\mathrm{E}_D(f(\mathbf{x}))$ and $\mathrm{Var}_D(f(\mathbf{x}))$ can be computed for any new $\mathbf{x}$ using Eqs. (A2) and (A3) in Appendix A1.3. Note that this calculation in-

cludes the adjusted expectation and variance of the $\rho_i$. Examining these quantities shows which regions and

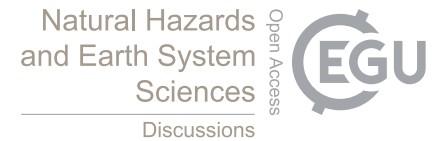

for which $g_i$ the differences between the fast and slow simulators are most pronounced. In conjunction with the $\beta_i'$, they also give more insight into how the active parameters drive the simulator output.

As before, the correlation lengths $\delta$ are tuned using leave-one-out validation. For the slow simulator, it was found that longer correlations lengths were more appropriate, with values ranging from $0.2$ to $0.4$. An example of the predictions of this validation, compared with the observed slow simulator output, can be found in Fig. 6. Over the 75 regions, the proportion of successful predictions from the validation ranged from $94.5\%$ to $99\%$.

For most emulators, the $\rho_i$ were close to 1 (typically between $0.95$ and $1.05$) for all $\beta_i$. With the difference between the fast and slow simulators being only a factor of $\sqrt{10}$ in the simulation noise and with the simulation noise being kept low by averaging over large regions, this is perhaps expected. The main exceptions were regions where the fast simulator predicted relatively little ash compared with the slow simulator—in these cases the $\rho_i$ were typically between $0.5$ and $0.75$ systematically (that is, no particular parameter was affected more than others). In no case did a $\rho_i$ approach $0$ (which would indicate a parameter becoming inactive in the slow emulator) or change sign. The only multiplier that was frequently low was $\rho_0$, the multiplier for the residual process. In conclusion, the link between mean functions of the two emulators is strong and consistent, in the sense that either the $\rho$ are all near $1$, or they are all near some $\alpha$ so that the difference is mostly a rescaling. The local variations, on the other hand, are usually unrelated, with $\rho_0$ near zero. This suggests that the fast simulator could be used more extensively in future applications significantly reducing simulation run times.

## 6.4 Implications for NAME case study

The adjusted $\beta_i$ confirm that the simulator behaves broadly as one would expect. As mass eruption rate increases (either due to its dependence on the plume rise height, $x_3$ via Eq. (1) or alterations in (1) caused by $x_3$) the quantity of ash in the atmosphere increases. When the precipitation threshold is higher, higher values of ash in the atmosphere are also predicted. This is due to less ash being deposited to the surface as only precipitation rates above the threshold lead to wet deposition. When the particle size distribution favours large particles, predicted airborne ash reduces because these heavy particles sediment much more





quickly than small particles and therefore are removed from the atmosphere and not available for long-range transport. The parameter $x_7$ governing free tropospheric turbulence is more interesting: low and high values lead to relatively less ash predicted than values towards the middle of the range. This is because at the extremes the ash has either been widely spread and diluted or it has not spread enough to reach the

region being considered in significant quantities. Note $x_7$ represents both horizontal and vertical turbulence because it is linked to $x_9$. Also becasue the effect in NAME is primarily through diffusivities $\sigma_u^2 \tau_u$ and $\sigma_w^2 \tau_w$, sensitivity to $x_7$ implies sensitivity to $\tau_u$ and $\tau_w$ too, although at a lower level.

Of all the parameters, the plume height drives the output most strongly, followed by the mass eruption rate and the precipitation threshold. In all cases, the $\beta_i$ with the highest adjusted expectation corresponded to a

function of the plume height, $x_1$ (either the $x_1$ term or the $x_1^2$ term). As a proportion of this, the adjusted expectations for $\beta_i$ corresponding to function of mass eruption rate, precipitation threshold, and turbulence were around $0.3$, $0.25$, and $0.1$ respectively. Adjusted expectations for $\beta_i$ corresponding to other parameters were typically lower than $0.1$ of that for plume height.

Interactions between the parameters (that is, the terms of the form $\beta_{ij} x_i x_j$) are mostly relatively small,

although each pair of $x_1$, $x_7$, and $x_{12}$ (plume height, turbulence in the free troposphere, and precipitation threshold respectively) have strong negative interactions. This means that, for example, although increasing plume height increases column loading, and increasing the precipitation threshold increases column loading, increasing both parameters at the same time does not increase column loading as much as would be expected looking only at the individual parameters.

The emulators provide insight into which areas of the parameter space will lead to high values of simulated ash column loading and which areas will lead to low values of ash column loading. As an extreme case, the parameters giving the lowest and highest predictions of ash column loading can be identified. This was done

5  for the first hour of 14 May, giving two parameter sets at which the simulator was evaluated. The results of these simulator evaluations can be seen in Fig. 7. This gives an idea of the range of possibilities admitted by the expert judgements from Sect. 3. As can be seen, these two plots are very different; that is, the ranges in Table 1 cover a broad range of simulator behaviour. Note however that our choice of parameter ranges has deliberately tried to cover the whole range of possible values and that, for some parameters such as those

10  relating to turbulence, it is not plausible that the extremes could be present throughout the simulation region.



Now, only a small region of this parameter space will lead to simulations that resemble the observations on this day. The emulators can be used to identify this region of parameter space. Since emulators can be evaluated very quickly, predictions and their associated uncertainty can be generated for very many candidate parameters, and all predictions that are very far from the observations can be rejected. This procedure, called *history matching*, focuses on the plausible regions of parameter space and allows more accurate emulators to be built within them. Performing this analysis for NAME is beyond the scope of this paper, but will be covered in a second study.

It is worth considering what advantages this analysis gives over a traditional one-at-a-time approach. There are three main benefits. The first is a quantitative assessment of the influence of changing each parameter to any new value. The second is an associated uncertainty for this assessment. The third is the treatment of interactions between parameters, which cannot be present in a one-at-a-time analysis. For instance, the approach used in this study can identify when there is a pair of parameters such that increasing either separately increases output, but increasing both simultaneously decreases output.

# 7 Conclusions

In this paper it has been shown that a Bayes linear emulation approach can be used to identify source and internal model parameters that contribute most to the uncertainty in the long-range transport of volcanic ash in a complex VATD simulator. The approach presented is applicable to other complex simulators that have long computation times and many parameters contributing to the overall prediction uncertainty. This approach uses latin hypercube sampling of the plausible parameter ranges determined through expert elicitation. All parameters are varied in each simulator run and therefore information about the importance of the parameters and their interaction can be investigated simultaneously. This gives a much more realistic estimate of the uncertainty than using one-at-a-time tests and provides much more useful information to model developers and those planning observational campaigns.

Here 1700 simulator runs have been used to build 75 emulators representing the average ash column loading in regions on 14 May 2010. These simulator evaluations comprised 1500 fast simulator runs and 200 slow simulator runs. The analysis demonstrated the strength of using approximate simulators to determine the



general trend of a simulator and provide plausible priors, before using a relatively small number of accurate simulator runs to refine the emulator. Bayes linear methods were used to reduce computational complexity and the need for detailed prior judgements that we may not believe.

For this case the most important parameters are plume height, mass eruption rate, free troposphere turbulence levels and precipitation threshold for wet deposition. There is also a strong negative relationship between each pair of the first three of these. These conclusions should be tested in other situations to assess how widely they hold. This information can be used to inform future research priorities (e.g. the addition of a more complex free tropospheric turbulence scheme which varies spatially and temporally (see Dacre et al. (2015)) and investigating the importance of the precipitation threshold within the NAME simulator) and observational capabilities (e.g. a mobile radar to observe plume height) and measurement campaigns (e.g. insitu observations of ash particle size distribution). Furthermore, this analysis can be used to prioritise variables to perturb in a small operational ensemble.

This study has shown the range of possible ash column loading distributions possible from sampling the parameter space determined by the ranges elicited from simulator experts. Only a small region of this parameter space will lead to simulations that resemble the observations on this day. Emulators can be used to identify this region of parameter space as they can be evaluated very quickly. The resulting predictions and their associated uncertainty can be generated for very many candidate parameters, and all predictions that are very far from the observations can be rejected. This procedure, known as *history matching*, focuses on the plausible regions and allows more accurate emulators to be built within them. This analysis is beyond the scope of this paper. This will form the basis of a future study but could further inform the parameter perturbations used in an operational ensemble. The approach presented here could be easily applied to other case studies, simulators or hazards. Furthermore, an ensemble of emulator evaluations could be used to produce probabilistic hazard forecasts.


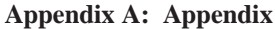


# Appendix A: Appendix

## A1 Adjusting slow emulators using fast emulators

### A1.1 Fitting the fast emulator

For the fast emulator, the $\beta'_i$ are fixed at their least squares estimates, and $\mathrm{Var}\left(u'(\mathbf{x})\right)$ is set to the residual standard deviation. This leaves only $\mathrm{Corr}\left(u'(\mathbf{x}_1), u'(\mathbf{x}_2)\right)$ to be determined. A typical approach is to specify a correlation function that depends only on $d(\mathbf{x}_1, \mathbf{x}_2)$, the distance between $\mathbf{x}_1$ and $\mathbf{x}_2$. A common choice, used in this study, is

$$\mathrm{Corr}\left(u'(\mathbf{x}_1), u'(\mathbf{x}_2)\right) = \exp\left(-\left(\frac{d(\mathbf{x}_1, \mathbf{x}_2)}{\delta'}\right)^2\right),$$

although other choices are possible; in particular using a different correlation length $\delta'$ for each direction would often be useful, although did not prove necessary in this application.

The parameter $\delta'$ governs the strength of the correlation, and must be estimated from the observed residuals by some method. A formal estimation can be performed using the variogram methods in, for instance, Cressie, 1993, applied, for instance, in Cumming and Goldstein, 2009. A more heuristic approach has been successful in many other applications (Vernon et al., 2010; Goldstein et al., 2010; Goldstein and Huntley, 2016). This involved the argument that, for a polynomial mean function, a plausible value of $\delta'$ is $\frac{1}{p+1}$ where $p$ is the highest-order term in the polynomial fit. This starting value can then be explored and adjusted "by hand". A popular strategy is a leave-one-out exploration: for each parameter $\mathbf{x}_i$, calculate the adjusted expectation and variance for $f(\mathbf{x}_i)$ using all the other $\mathbf{x}_j$ and a trial value of $\delta'$. The observed value of $f(\mathbf{x}_i)$ can then be used to see whether the prediction was accurate or not. The value of $\delta'$ used can be adjusted to balance two competing requirements: that most of the predictions are close (relative to the adjusted variance) to the observed values, and that the variances are small. For example, if many more than 5% of predictions are more than three standard deviations away from the observation, $\delta'$ is unlikely to be a good choice, so a good value of $\delta'$ should satisfy this requirement while keeping the variances as low as possible.





## A1.2 Prior judgements for linked emulators

Using the linking model in Eq. (12), an adjustment of the slow emulator involves prior expectations, variances, and covariances for $\rho_i$, $\rho_0$, and $w(\mathbf{x})$. A simple approach is to use

$$\mathrm{E}\left(\rho_i\right) = 1$$
$$\mathrm{Var}\left(\rho_i\right) = \sigma_\rho^2$$
$$\mathrm{Cov}\left(\rho_i, \rho_j\right) = r,$$

reducing the specification for the multiplier to two numbers $\sigma_\rho^2$ and $r$. Note that $\rho_0$ is included in the above specifications. This leaves only $w(\mathbf{x})$ to consider. A natural choice is to use the same form as is used for

20   $u'(\mathbf{x})$, including the same variance and the same correlation length $\delta$. Another option is to use the same correlation structure, but allow $\mathrm{Var}\left(w(\mathbf{x})\right) = \sigma_w^2$ to be different from $\mathrm{Var}\left(u'(\mathbf{x})\right)$. Finally, a very useful simplification is to take $\mathrm{Corr}\left(w(\mathbf{x}), \rho_i\right) = 0$ for all $i$ (including 0).

Thus, the link between $f'$ and $f$ is provided by $\tau = \{\sigma_\rho^2, r, \sigma_w^2\}$—only these three values need to be specified now (or only the first two, depending on earlier choices). Were it possible to specify values for $\tau$, this would provide all the ingredients to perform a Bayes linear calculation to learn about the slow simulator using the (adjusted) fast emulator and the evaluations $f(\mathcal{X}_S)$. However, the quantities in $\tau$ are difficult to think about, so expecting an expert to be able to specify them is unrealistic.

Instead, plausible values for $\tau$ can be generated using the differences $d(\mathbf{x}) = f(\mathbf{x}) - f'(\mathbf{x})$ for each $\mathbf{x} \in \mathcal{X}_S$.

As calculated in Cumming and Goldstein, 2009,

$$\mathrm{Var}\left(d(\mathbf{x})\right) = \sigma_\rho^2 \phi(\mathbf{x}) + \sigma_\rho^2 r \psi(\mathbf{x}) + \sigma_w^2, \tag{A1}$$

where

$$\phi(\mathbf{x}) = \sum_{i=1}^{p+1} b_i(\mathbf{x})^2$$
$$\psi(\mathbf{x}) = \sum_{i \neq j} b_i(\mathbf{x}) b_j(\mathbf{x}),$$

with

$$b(\mathbf{x}) = (\beta_1' g_1(\mathbf{x}), \dots, \beta_p'(\mathbf{x}) g_p(\mathbf{x}), u'(\mathbf{x})),$$


noting that $u'(\mathbf{x})$ is known for each $\mathbf{x} \in \mathcal{X}_S$ because the fast simulator was evaluated at each such point. Further, $\mathrm{E}\left(d(\mathbf{x})\right) = 0$, and hence $\mathrm{Var}\left(d(\mathbf{x})\right) = \mathrm{E}\left(d(\mathbf{x})^2\right)$ and so from Eq. (A1),

$$\mathrm{E}\left(d(\mathbf{x})^2\right) = \sigma_\rho^2 \phi(\mathbf{x}) + \sigma_\rho^2 r \psi(\mathbf{x}) + \sigma_w^2,$$

and for $\mathbf{x} \in \mathcal{X}_S$ everything on the right-hand side of this equation is known except for $\tau$. Replacing $\mathrm{E}\left(d(\mathbf{x})^2\right)$ with the observed $d(\mathbf{x})^2$, this gives $|\mathcal{X}_S|$ (in our application, 200) linear equations in 3 unknowns, and a least-squares fit can be used to estimate these three unknowns and hence $\hat{\tau}$. This $\hat{\tau}$ can then be used as the prior judgements for the link between the emulators. Note that this approach works only because both fast and slow simulators are evaluated at $\mathcal{X}_S$.

## A1.3    Adjusting the slow emulator

Suppose an emulator $f'$ has been constructed as in Eq. (11) by using $D'$; in particular we suppose that the $beta'_i$ are known and that $u'$ has had its mean and variance adjusted using (9) and (10) (with $D$ replaced by $D'$). We also assume the link (12) between the fast and slow emulators and that priors have been specified for $\rho_i$ and $w$, for instance by the methods in Appendix A1.2. The adjusted fast emulator and the slow simulator runs $D$ are available to be used in the adjustment of $\rho$ and $w$, and hence the adjustment of $f(\mathbf{x})$ for any new $\mathbf{x}$.

5    First, we have

$$\mathrm{E}_D(\rho_i) = 1 + \mathrm{Cov}\left(\rho_i, D\right)\mathrm{Var}\left(D\right)^{-1}\left(D - \mathrm{E}(D)\right).$$

The prior expectation for each element of $D$ is simply the value observed for the corresponding element of $D'$. Also,

$$
\begin{aligned}
\mathrm{Cov}\left(\rho_i, D_j\right) &= \mathrm{Cov}\left(\rho_i, \sum_k \rho_k \beta'_k g_k(\mathbf{x}_j) + \rho_0 u'(\mathbf{x}_j) + w(\mathbf{x}_j)\right) \\
&= \sum_k \mathrm{Cov}\left(\rho_i, \rho_k\right)\beta'_k g_k(\mathbf{x}_j) + \mathrm{Cov}\left(\rho_i, \rho_0\right) u'(\mathbf{x}_j) \\
&= \sum_{k=1}^{p+1} \mathrm{Cov}\left(\rho_i, \rho_k\right) b(\mathbf{x}_j)_k.
\end{aligned}
$$




Finally, the variance matrix $\mathrm{Var}\,(D)$ is built from elements of the form

$$
\mathrm{Cov}\,(D_1, D_2) = \mathrm{Cov}\left(\sum_i \rho_i \beta_i' g_i(\mathbf{x}_1) + \rho_0 u'(\mathbf{x}_1) + w(\mathbf{x}_1), \sum_i \rho_i \beta_i' g_i(\mathbf{x}_2) + \rho_0 u'(\mathbf{x}_2) + w(\mathbf{x}_2)\right)
$$
$$
= \sum_{i,j} \mathrm{Cov}\,(\rho_i, \rho_j)\, b(\mathbf{x}_1) b(\mathbf{x}_2) + \mathrm{Cov}\,(w(\mathbf{x}_1), w(\mathbf{x}_2)).
$$

15   This is all that is needed to calculate $\mathrm{E}_D(\rho_i)$.

The adjusted variance for $\rho$ is given by

$$
\mathrm{Var}_D(\rho) = \mathrm{Var}\,(\rho) - \mathrm{Cov}\,(\rho, D)\,\mathrm{Var}\,(D)^{-1}\,\mathrm{Cov}\,(D, \rho),
$$

which can be calculated from the expressions above.

The adjustment for the residual $w(\mathbf{x})$ is simpler:

20   $\mathrm{E}_D(w(\mathbf{x})) = \mathrm{Cov}\,(w(\mathbf{x}), D)\,\mathrm{Var}\,(D)^{-1}\,(D - \mathrm{E}\,(D))$

$\mathrm{Var}_D(w(\mathbf{x})) = \mathrm{Var}\,(w(\mathbf{x})) - \mathrm{Cov}\,(w(\mathbf{x}), D)\,\mathrm{Var}\,(D)^{-1}\,\mathrm{Cov}\,(D, w(\mathbf{x}))$

where

$$
\mathrm{Cov}\,(w(\mathbf{x}), D_i) = \mathrm{Cov}\,(w(\mathbf{x}), w(\mathbf{x}_i)).
$$

Then, for any $\mathbf{x}$ such that $\mathbf{x} \in \mathcal{X}_F$ (the parameters used for the fast but not slow simulator runs), Cumming and Goldstein,

5   2009 showed that the Bayes linear adjustment for $f(\mathbf{x})$ is given by

$$
\mathrm{E}_D(f(\mathbf{x})) = b(\mathbf{x})^T \mathrm{E}_D(\rho) + \mathrm{E}_D(w(\mathbf{x})) \tag{A2}
$$

$$
\mathrm{Var}_D(f(\mathbf{x})) = b(\mathbf{x})^T \mathrm{Var}_D(\rho) b(\mathbf{x}) + \mathrm{Var}_D(w(\mathbf{x})) + 2b(\mathbf{x}) \mathrm{Cov}_D(\rho, w(\mathbf{x})), \tag{A3}
$$

where

$$
\mathrm{Cov}_D(\rho, w(\mathbf{x})) = \mathrm{Cov}\,(\rho, w(\mathbf{x})) - \mathrm{Cov}\,(\rho, D)\,\mathrm{Var}\,(D)^{-1}\,\mathrm{Cov}\,(D, w(\mathbf{x}))
$$
$$
= -\mathrm{Cov}\,(\rho, D)\,\mathrm{Var}\,(D)^{-1}\,\mathrm{Cov}\,(D, w(\mathbf{x})),
$$

recalling that $\mathrm{Cov}\,(\rho, w(\mathbf{x}))$ was assumed to be zero.

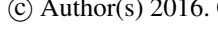



For new $\mathbf{x}$ for which the fast simulator has not been evaluated, the equations remain almost identical, but

there is the added complication that $u'(\mathbf{x})$, the residual in the fast emulator, is not known. Since this appears

in the final element of $b(\mathbf{x})$, the above equations cannot be evaluated. Under the assumption that the slow

simulator runs $D$ provide no further information about the fast simulator, the final element of $b(\mathbf{x})$ in these

equations can be treated as fixed at the adjusted expectation $\mathrm{E}_{D'}(u'(\mathbf{x}))$.

*Acknowledgements.* We thank Andy Hart from Food and Environment Research Agency for useful discussions and

helping us to conduct the expert elicitation. Natalie Harvey and Nathan Huntley gratefully acknowledge funding from

NERC grant NE/J01721/1 Probability, Uncertainty and Risk in the Environment.

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

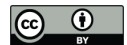

| Key | Parameter name | Default value | Minimum value | Maximum value |
|---|---|---|---|---|
| $x_1$ | Height of plume at release (m) | Taken from Arason et al. (2011) | default $-$ 2000m | default $+$ 2000m |
| $x_2$ | Thin layer depth (m) | 1000m | 100m | 2000m |
| $x_3$ | Mass eruption rate (kg s$^{-1}$) | As per Mastin et al. (2009) | default/3 | default $\times$ 3 |
| $x_4$ | Shape parameter for the Gamma distribution for particle sizes | NA | 3 | 10 |
| $x_5$ | Scale parameter for the Gamma distribution for particle sizes ($\mu$m) | NA | 1 | 10 |
| $x_6$ | Density of the ash (kg m$^{-3}$) | 2300 | 1350 | 2700 |
| $x_7$ | Standard deviation of horizontal velocity for free tropospheric turbulence (m s$^{-1}$). Varied in proportion to $x_8$. | 0.25 | 0.0025 | 2.5 |
| $x_8$ | Standard deviation of vertical velocity for free tropospheric turbulence (m s$^{-1}$). Varied in proportion to $x_7$. | 0.1 | 0.001 | 1 |
| $x_9$ | Horizontal Lagrangian timescale for free tropospheric turbulence (s). Varied in proportion to $x_{10}$. | 300 | 100 | 900 |
| $x_{10}$ | Vertical Lagrangian timescale for free tropospheric turbulence (s). Varied in proportion to $x_9$. | 100 | 20 | 300 |
| $x_{11}$ | Standard deviation of horizontal velocity for unresolved mesoscale motions (m s$^{-1}$) | 0.8 | 0.27 | 1.74 |
| $x_{12}$ | Precipitation rate required for wet deposition to occur (mm hr$^{-1}$) | 0.03 | 0 | 0.1 |
| $x_{13}$ | Scavenging coefficient parameter A for rain (s$^{-1}$) | Below cloud: $8.4\times10^{-5}$ | 0.000001 | 0.01 |
| | | In cloud: $3.36\times10^{-4}$ | | |
| $x_{14}$ | Scavenging coefficient parameter A for snow (s$^{-1}$) | Below cloud: $8.0\times10^{-5}$ | 0.000001 | 0.1 |
| | | In cloud: $5.2\times10^{-5}$ | | |
| $x_{15}$ | Scavenging coefficient parameter B for rain | 0.790 | 0.4 | 1.1 |
| $x_{16}$ | Scavenging coefficient parameter B for snow | Below cloud: 0.305 | 0.2 | 1.2 |
| | | In cloud: 0.790 | | |
| $x_{17}$ | Dry deposition aerodynamic resistance perturbation factor | 1 | 0.5 | 2 |
| $x_{18}$ | Dry deposition Laminar sublayer resistance numerator | 300 | 0 | 300 |

**Table 1.** Summary of the parameters, default values and ranges used in this study.



| Particle Diameter ($\mu m$) | Mass Fraction |
|:---:|:---:|
| 0.1 - 0.3 | 0.001 |
| 0.3 - 1.0 | 0.005 |
| 1.0 - 3.0 | 0.05 |
| 3.0 - 10.0 | 0.2 |
| 10.0 - 30.0 | 0.7 |
| 30.0 - 100.0 | 0.044 |

**Table 2.** The default input source PSD used in NAME by the London VAAC.





| | First region | Second region | Third region | Fourth region |
|---|---|---|---|---|
| 0000 UTC | $(-13, 61):(-5, 69)$ | $(-13, 55):(-6, 61)$ | $(-22, 59):(-13, 65)$ | |
| 0100 UTC | $(-14, 62):(-6, 69)$ | $(-14, 55):(-6, 62)$ | $(-22, 60):(-14, 65)$ | |
| 0200 UTC | $(-14, 61):(-6, 69)$ | $(-14, 54):(-6, 61)$ | | |
| 0300 UTC | $(-14.5, 61.5):(-6.5, 69.5)$ | $(-14.5, 54):(-4, 61.5)$ | | |
| 0400 UTC | $(-15, 62):(-6, 70)$ | $(-15, 54):(-5, 62)$ | | |
| 0500 UTC | $(-15.5, 61):(-6, 70)$ | $(-15, 53):(-3, 61)$ | | |
| 0600 UTC | $(-15.5, 61):(-6, 70)$ | $(-15, 53):(-3, 61)$ | | |
| 0700 UTC | $(-17, 63.5):(-9, 70)$ | $(-14.5, 59):(-6, 63.5)$ | $(-11, 53):(-2, 59.5)$ | |
| 0800 UTC | $(-18, 64):(-9, 70)$ | $(-15, 61):(-8, 64)$ | $(-11, 53):(-1, 61)$ | $(-27, 63):(-19, 66)$ |
| 0900 UTC | $(-20.5, 64):(-9, 71)$ | $(-15, 61):(-8, 64)$ | $(-11, 53):(-1, 61)$ | $(-28, 63):(-20, 66)$ |
| 1000 UTC | $(-21, 64.5):(-9, 71)$ | $(-15, 61):(-8, 64.5)$ | $(-11, 53):(-1, 61)$ | $(-30, 63):(-21, 66)$ |
| 1100 UTC | $(-21, 63):(-9, 71)$ | $(-12, 53):(-1, 62)$ | $(-30, 63):(-21, 66)$ | |
| 1200 UTC | $(-22, 63.5):(-9, 71)$ | $(-12, 53):(-1, 62)$ | $(-31, 63):(-23, 66)$ | |
| 1300 UTC | $(-23, 63):(-10, 71)$ | $(-12, 53):(-1, 62)$ | $(-32, 63):(-23, 66)$ | |
| 1400 UTC | $(-24, 65):(-17, 71)$ | $(-17, 63):(-12, 67)$ | $(-12, 52):(0, 62)$ | $(-33, 62.5):(-22, 66.5)$ |
| 1500 UTC | $(-24, 65):(-18, 71)$ | $(-18, 63):(-12, 67)$ | $(-8, 53):(0, 59)$ | $(-33, 62.5):(-22, 65.5)$ |
| 1600 UTC | $(-25, 64):(-20, 71)$ | $(-20, 62):(-12, 66)$ | $(-8, 52):(0, 58)$ | $(-33, 62.5):(-24, 66)$ |
| 1700 UTC | $(-26, 65):(-19, 71)$ | $(-20, 62):(-15, 65)$ | $(-8, 52):(0, 58)$ | $(-34, 62.5):(-24, 66)$ |
| 1800 UTC | $(-28, 66):(-19, 71)$ | $(-27, 62):(-14, 66)$ | $(-7, 52):(1, 58)$ | $(-34, 62.5):(-27, 66)$ |
| 1900 UTC | $(-27, 62):(-14, 67)$ | $(-7, 52):(1, 57)$ | $(-34, 62.5):(-27, 66)$ | |
| 2000 UTC | $(-27, 62):(-14, 67)$ | $(-7, 52):(1, 57)$ | $(-36, 62.5):(-27, 66.5)$ | |
| 2100 UTC | $(-27.5, 61.5):(-18, 67)$ | $(-7, 52):(1, 57)$ | $(-37, 62):(-27.5, 66.5)$ | |
| 2200 UTC | $(-28, 63.5):(-18, 67)$ | $(-7, 51.5):(1, 55.5)$ | $(-37, 62):(-28, 66.5)$ | |
| 2300 UTC | $(-30, 63.5):(-18, 66.5)$ | $(-7, 51.5):(1, 55.5)$ | $(-37, 62):(-30, 66.5)$ | |

**Table 3.** Location of geographical regions used for comparision for each hour by longitude and latitude of the region corners.



| Parameter | $x_1$ | $x_3$ | $x_4$ | $x_5$ | $x_6$ | $x_7$ | $x_9$ | $x_{11}$ | $x_{12}$ | $x_{13}$ | $x_{14}$ | $x_{15}$ | $x_{16}$ | $x_{17}$ | $x_{18}$ |
|---|---|---|---|---|---|---|---|---|---|---|---|---|---|---|---|
| Times active | 75 | 75 | 18 | 18 | 0 | 61 | 15 | 4 | 58 | 0 | 1 | 0 | 0 | 0 | 0 |

**Table 4.** Number of outputs for which each parameter was judged active (and hence included in the emulator for that output). Recall that $x_7$ and $x_8$ are linked, and so $x_8$ is not present in the table, and similarly for $x_{10}$ which is linked to $x_9$.


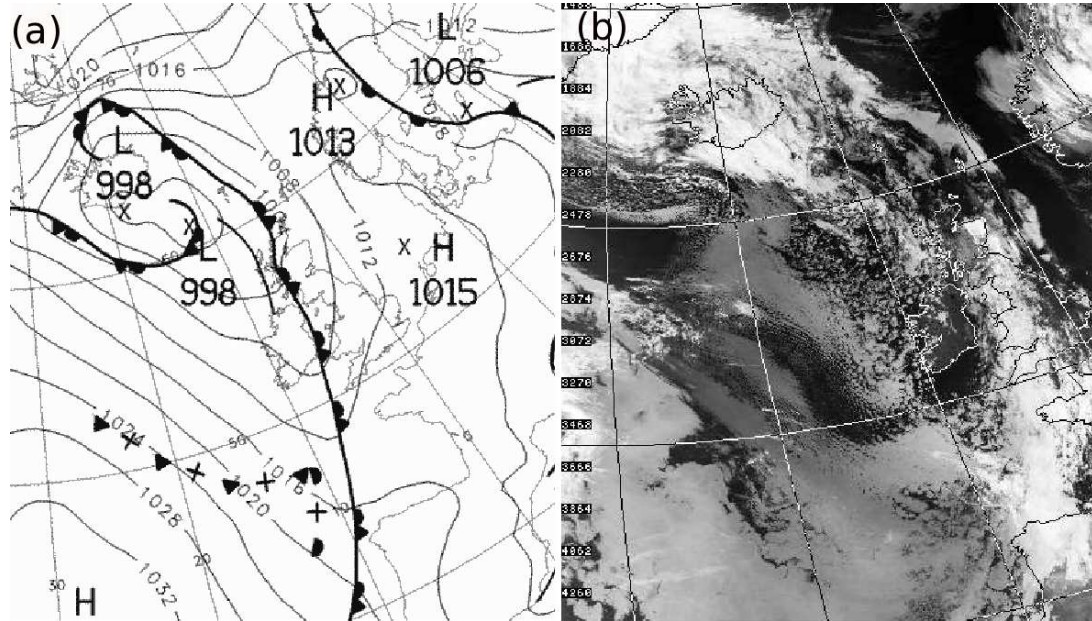

**Figure 1.** (a) UK Met Office surface analysis chart at 0000 UTC on 14 May 2010. Mean sea level pressure isobars overlaid with surface fronts.(b) AVHRR infrared satellite image at 0613 UTC on the 14 May 2010 provided by the Dundee satellite receiving station.





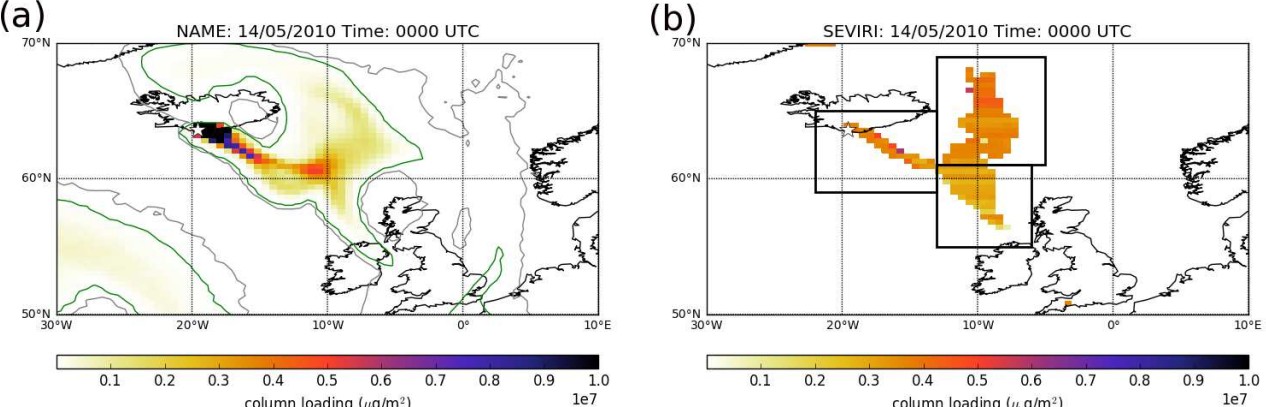

**Figure 2.** (a) Simulated ash column loading at 0000 UTC 14 May 2010 using parameters near the default values. (b) SEVIRI satellite retrieved ash column loading also at 0000 UTC 14 May 2010. The black boxes denote the regions over which average ash column loading is being emulated for this hour. In (a) column loading of 20000 $\mu$g/m$^2$ and 2000 $\mu$g/m$^2$ are shown by the green and grey contours respectively.

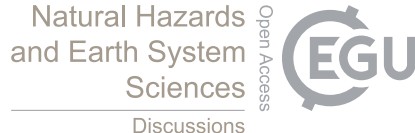

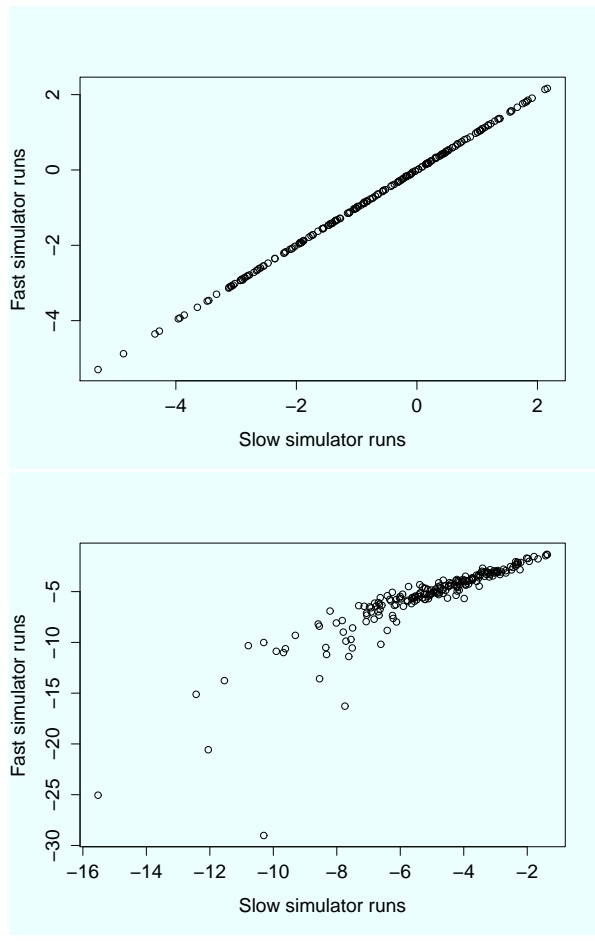

**Figure 3.** Relationship between the slow simulator and fast simulator output for $\mathcal{X}_S$ at (a) the first region and (b) the 63rd region (third region at 1900 UTC). The 63rd region has the lowest correlation between fast and slow simulator output.



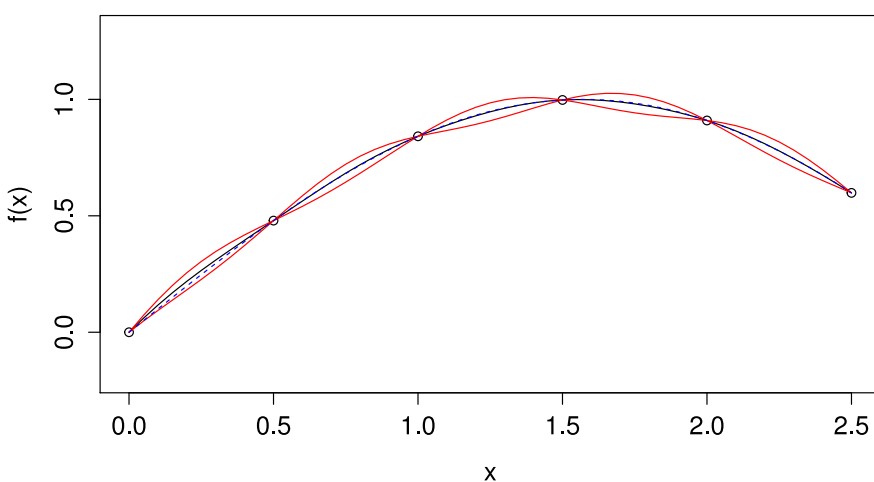

**Figure 4.** One-dimensional example of an emulator. The points represent the six evaluations of $f(\mathbf{x})$, the black line is the emulator's prediction, and the red lines give two standard deviations. The blue dashed line is the true value of $f(\mathbf{x})$.

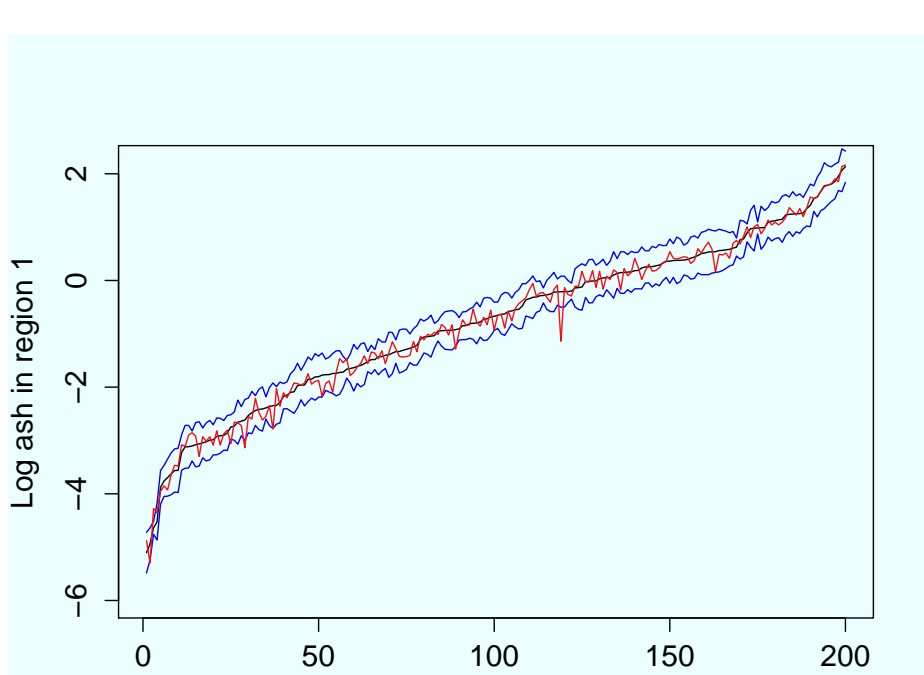

**Figure 5.** Validation plot for the emulator of the first output. Emulator expected value for the parameter sets in $\mathcal{X}_S$ is shown in black, with an interval of three standard deviations each side shown in blue. The red line shows the true simulator output at each parameter set. The parameters have been ordered from lowest to highest emulator prediction.





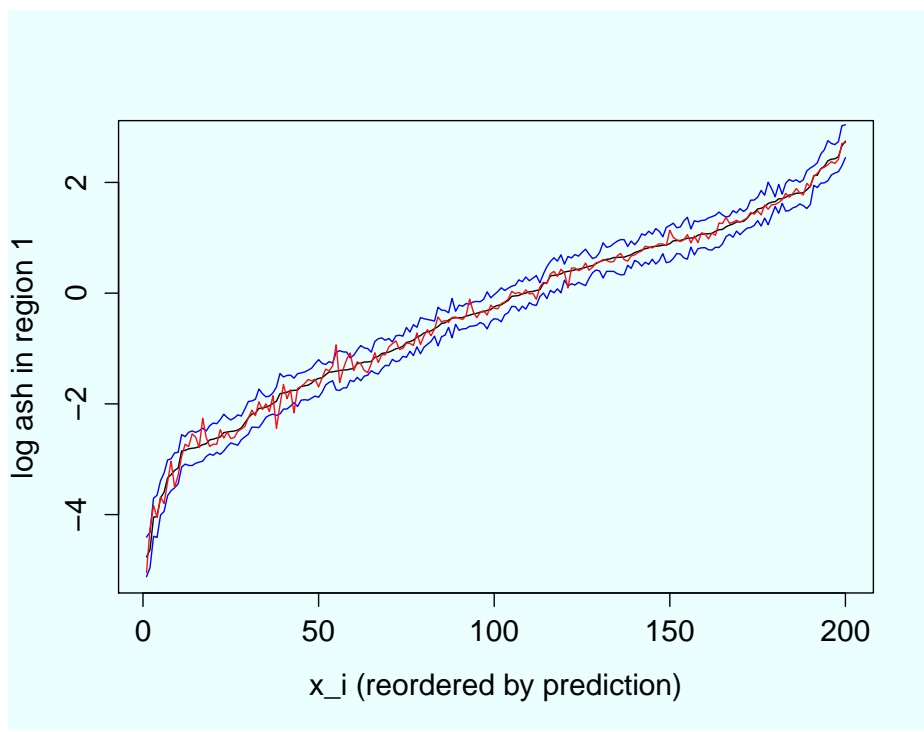

**Figure 6.** Leave-one-out validation plot for the emulator of the slow simulator. Emulator expected value is shown in black, with an interval of three standard deviations each side shown in blue. The red line shows the true simulator output at each parameter set. The parameters have been ordered from lowest to highest emulator prediction.




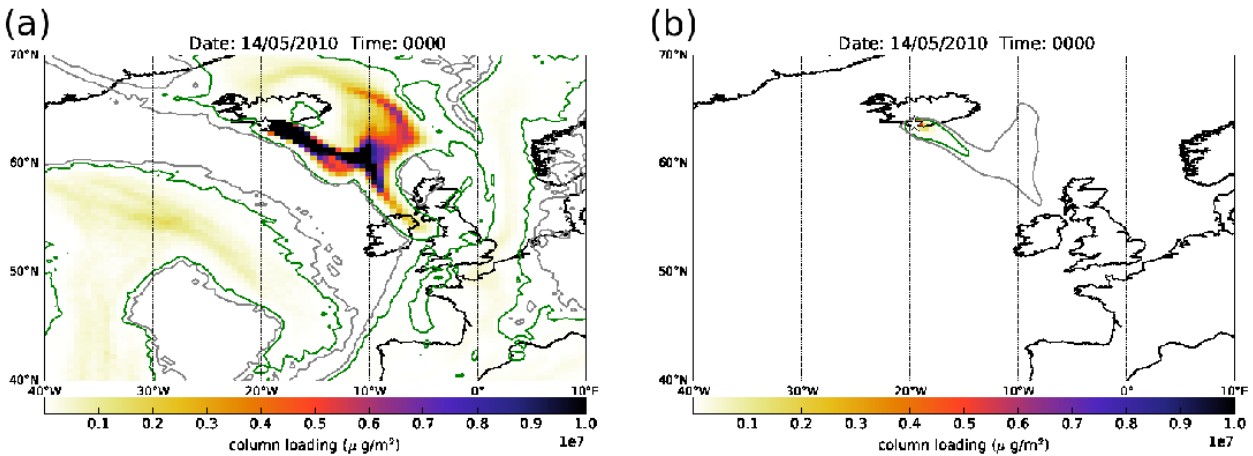

**Figure 7.** NAME ash column loading for parameter choices with the highest and lowest expected ash column loadings in the first geographical region at 0000 UTC 14 May. The contours are as in Figure 2.