# Peer review of "Multi-level emulation of a volcanic ash transport and dispersion model to quantify sensitivity to uncertain parameters"

_Natural Hazards and Earth System Sciences, 2016_

## Referee Comment (RC1) · Anonymous Referee #1 · 15 Oct 2016

The subject manuscript describes an inversion of ash cloud transport in order to make a forecast. The manuscript is poorly written and organized, and is hard to read critically. It has one glaring weakness in that Bayesian linear regression is used on a high dimension problem formulated by the authors. Yet the emulator operates in a way to just constrain those parts of the model that are constrained by the data. This is a desirable property, but carries with it a liability. With this multi-level approach, there is the possibility (very likely) that there is a highly nonlinear dependence on the network function of the parameter values, in which case an exact Bayesian treatment is no longer applicable. The posteriors may be multi-model, and possibly non-convex. None of this is addressed, and it is also very difficult to follow exactly how the model operated on the

[Figure]

May 14, 2010 ash clouds from Eyjafjallajokull volcano to provide the results in Table 4. My recommendation is that the manuscript be completely reorganized and rewritten, being more explicit in some areas but also putting some explanations in Appendices or deferring to previously published work. It is very difficult to assess the efficacy and veracity of this study. Yet this is an important problem that deserves better treatment than is presented here. This could have been a shorter and much more informative paper than it is in its present state.

Specific comments: 1) Use references with descriptions of how NAME works to shorten section 2. 2) page 5, line 24: particle density and particle size distribution are not known at the time of an eruption, so a forecast model must rely on a priori information. 3) page 13, line 17: only fine particles are considered, so why have particle size distribution as a variable, increasing the dimensionality, in this model (described in a later section). 4) page 14, line 20: 10,000 per hour, 1000 per hour are the rates of particle release, not the number of particles. 5) page 16, lines 15-20: these are properties. So remove the word 'must' and replace 'should predict' with 'requires'. 6) page 16: write out the function g, here equal to 'x' to the ith power. 7) page 16: define both expectation and correlation rather than just putting in E() and Corr() and expecting the reader to just know what this means. 8) page 17: same comment as above for Var() showing up suddenly in the text. 9) page 17: line 25: 'some of these problems are summarized'. Nice lead in, except that none of these problems are summarized later on. I suspect they got abandoned in an earlier draft. 10) 'expert judgements' throughout are the a priori in your Bayesian analysis. 11) page 18: The paragraph/sentence starting with "Such an approach" is a non-sequitor with the paragraph and sentence immediately above. What approach? 12) page 18, 1st paragraph: forecasting uses the posterior, which has the prior in it. Change 'For calibration and forecasting' to 'For calibration' 13) page 18: The reason that doing high dimension problems with a linear Bayesian analysis is very difficult is that linear regression is not well suited to high dimension problems. You don't have enough flexibility with fixed basis functions. This could have been alleviated in this paper by just considering two parameters for distal clouds; height and mass

flux at that height. 14) page 20, first paragraph: so, a prior was not used here? 15) page 22: give references for R squared. 16) page 24, line 24: change "..., whereas of course" to "..., even though". 17) page 24, bottom: so, the number of parameters could not be reduced below 4? What if you did not do this? Would you just end up with height and mass flux? I suspect so. 18) page 34, Cov equations at top: there are 2 parameter sets here. Need either references or a proof of the second equation.
* * *

---

## Referee Comment (RC2) · F. Pianosi (Referee) · 18 Oct 2016

The manuscript presents a complex and potentially very interesting application of emulation modelling to quantify the relative impact of several uncertain parameters on the predictions of a volcanic ash transport and dispersion model (the NAME model). The main novelty of this study is that it presents one of the first applications of a formal sensitivity analysis (i.e. beyond one-at-the-time approaches) to a volcanic ash transport and dispersion model. Moreover, some of the techniques and lessons learnt in this study (for example on how to link fast and slow simulators) might be of interest to a broader community who deals with uncertainty in computationally expensive transport and dispersion models, not only volcanic ashes.

[Figure]

However, the manuscript in its present form is quite unclear and the structure unbalanced, which makes it difficult to fully evaluate and appreciate the findings of the study. Below my main issues and some suggestions for revision.

[1] The choice of contents in the methodology sections is confusing. Too much space is given to topics that are generic and covered in many other papers or even textbooks, which divert attention from and leave too little space to specific information about the NAME application. For example, on P. 14 L. 16-29 almost the same space is devoted to describing the difference between fast and slow simulators, which is a very important setting of the model under study (see also comment [3] below), as to describing the well-established and totally generic Latin Hypercube sampling technique. The description of emulators and Bayes linear methods covers 4 pages (P. 16-19) although much of the content is very generic, easily accessible elsewhere, and - most importantly - not strictly related to the application here. In fact, from P. 20 L. 10-12 ("a successful method has been to use a standard (non-Bayesian) least-squares regression") I understand that the non-linear/linear Bayesian approaches discussed on previous pages are not actually used here because linear least-squares are sufficient. As a reader I was confused by these long descriptions and diverted from focusing on the specific features of this application, for example the link between emulators of the fast and slow simulators. Another example is Section 5.4: this is all generic and standard methods that do not need to be discussed in an application-oriented paper, especially if not used then in the Results section (see for example the comment on "examining the residuals"...). In summary, I would recommend to deeply revise these sections, to make them shorter and include only the information relevant to the specific application and thus needed to understand the results.

[2] The results section seems to focus a lot on the validation of the emulators - i.e. how good they are in representing the fast and slow simulators - and their similarity, rather than the ultimate goal of the analysis, which is to use the emulators "as a research tool to better understand the simulator, the role of the parameters, the interactions between

them..." (P. 4 L. 17). For example, while Fig. 5 and 6 report validation results for the emulators, there are no Figures reporting sensitivity results, parameter mapping or interactions. The only results related to sensitivity analysis are in Table 4, which however does not even use the word "sensitivity"! This is odd especially considering the title of the manuscript.

[3] The difference between fast and slow simulators should be better clarified. In particular, on P. 14, L. 16-24: What does the "increase in particle-sampling noise" mean in simpler terms? How does it impact the simulation results? Not so much - at least for predicting average column loadings - according to what the authors say later on P. 15 (L. 7-12). On the other hand, how fast is the fast simulator? These aspects need to be clarified because they are at the basis of the motivation of the study: if the fast simulator is fast enough, and its predictions of the output of interest (average column loading) are reasonably close to those of the original (slow) simulator, then why building an emulator? One could directly apply a Global Sensitivity Analysis technique (for instance, the Morris method, or Regional Sensitivity Analysis, which are both reasonably "low-cost") to the fast simulator. I am not saying that this is necessarily the case (maybe the fast simulator is not so fast, or there is some other reason I am missing to avoid using it) but more details should be provided to clarify this crucial point.

[4] Choices underpinning the analysis and their impact on sensitivity results needs further discussion. P. 6, L. 20 onwards: there are three set of choices that are made here: - the choice of the uncertain parameters to be included in the sensitivity analysis (out of a larger set of parameters appearing in NAME, which are set to default values - two of them are further described in Sec. 3.2.7 and 3.2.8, together with the reasons for not varying them - what about the others?) - the choice of plausible ranges for the uncertain parameters - the choice of a particular event, and hence a particular set of forcing data, for the model simulation (and the choice of ignoring the uncertainty in those data) I presume that these choices may have a very strong impact on the results and thus on the generality/transferability of the findings. Assessing such impact might
be beyond the scope of this manuscript, but the point should be at least mentioned and discussed.

OTHER SPECIFIC POINTS:

P. 3 L. 28: "Finally, the analysis cannot provide ...". A bit vague, please clarify what is an "overall assessment of uncertainty"?

P. 8 L. 3-4: Difference between "full depth" and "thin layer" is not clear to me. Also, very unclear how the 1700 + 1700 runs here mentioned are connected with the 1500 + 200 runs mentioned on P. 14 (the same experiment of P. 14 is repeated twice, once per each source type?). Maybe the confusion could be avoided by simply not giving all these details on the thin layer source case, since its results are not shown (as commented on P. 23 L. 15-16)?

P. 10 L. 18: " are varied by the same proportion". Unclear.

P. 12 L. 23: "between 0 and 2": or between 0.5 and 2 (assuming this section refers to $x_{17}$ in Table 1).

P. 14, L. 10-11: "the average ash column loading predicted ..." not completely clear. Are column loadings per each region averaged over the simulation period, thus defining 75 "outputs" (and 75 emulators), or are predictions for each hour analysed separately (thus defining 75xT "outputs", where T is the number of hours in the simulation)? Please clarify, maybe also inserting an equation here.

P. 14, L. 14: Again unclear: "regions used for the first hour are marked..." So, the definition of regions changes from one hour to another? And also their number then? And so how does it connect to my previous question?

P. 15, L. 7-12 and Figure 3. Again on the difference between fast and slow simulators. I understand that the main conclusion here is that simulations from fast and slow simulators are similar, however this paragraph and the Figure are rather unclear. "agreement between the two simulators" means that they provide similar predictions of average

column loading? What does "to be related but not in agreement" mean? The "correlations" of 0.99 and 0.7 are the correlation between simulated column loadings (in different regions? at different time in the simulation period?). Please be more specific. Figure 3: units of measurements on the axes are missing!!! (I guess they are "Log ash" as in Fig. 5?).

P. 15, L. 7: "the goal" ... This is a bit misleading: the ultimate reason for building the emulator is not making inference, but rather understanding the role of parameters (see discussion on P. 4 L. 14-17). Maybe good to remind it here.

P. 28, L. 9-10: "for some parameters... it is not plausible": so you included in the analysis some parameter combinations that are implausible? This sounds contradictory (who would be interested in sensitivity to parameter variations that are not plausible?). Please clarify

P. 29, L. 18-23: This is a generic comment that was already made in the Introduction (P. 3) and is not part of the results. I would remove it.

Table 1: - term "default" on first row is a bit misleading - does it refer to the "data from Keflavik radar" (as explained on P. 7, L. 19)? If so, clarify in the Table - term "default" on third row is a bit misleading - does it refer to the MER value from Eq. (1) (as explained on P. 8, L. 20)? If so, clarify in the Table - connect better to the text, for example the mathematical symbols used in the text could be included in the description of "Parameter name" (for example R_a on row 17...) - rows 7-10: "varied in proportion..." is unclear (here and in the text) - row 18: so the default value coincides with the maximum value? This looks strange.

Figure 3: Caption mention "(a)" and "(b)" but letters are not reported in the panels.

[Figure]

---

## Author Comment (AC1) · 17 Jan 2017

Response to Reviewers for Manuscript nhess-2016-288: "Multi-level emulation of a volcanic ash transport and dispersion model to quantify sensitivity to uncertain parameters"

Editor/reviewer comments are black. Responses to comments by the authors are red.

To address the reviews, some major restructuring and expansions are necessary. We observe in the two reviews that one reviewer asks us to provide more detail in some places, particularly about some aspects of the methodology, whereas the second review would prefer some of this discussion to be shortened where appropriate. To address both of these requests adequately, it should be possible to significantly tighten the main account, but rather than omitting some of the detail, instead move the some text to the Appendix.

**Review 1**

We thank the reviewer for taking the time to review our manuscript.

The subject manuscript describes an inversion of ash cloud transport in order to make a forecast. The manuscript is poorly written and organized, and is hard to read critically. It has one glaring weakness in that Bayesian linear regression is used on a high dimension problem formulated by the authors. Yet the emulator operates in a way to just constrain those parts of the model that are constrained by the data. This is a desirable property, but carries with it a liability. With this multi-level approach, there is the possibility (very likely) that there is a highly nonlinear dependence on the network function of the parameter values, in which case an exact Bayesian treatment is no longer applicable. The posteriors may be multi-model, and possibly non-convex. None of this is addressed, and it is also very difficult to follow exactly how the model operated on the May 14, 2010 ash clouds from Eyjafjallajokull volcano to provide the results in Table 4.

My recommendation is that the manuscript be completely reorganized and rewritten, being more explicit in some areas but also putting some explanations in Appendices or deferring to previously published work. It is very difficult to assess the efficacy and veracity of this study. Yet this is an important problem that deserves better treatment than is presented here. This could have been a shorter and much more informative paper than it is in its present state.

The linear regression component is only part of the analysis. When the regression component is weak, then modelling the residual process would become much more important and would become the main focus of the analysis. The validation steps would typically alert us to these cases. In our application, the emulators validated well. It is true, however, that there will be parts of the parameter space where the regression has high influence but we have no samples nearby to test performance. If we were to identify these regions, we can run the simulator at these locations and perform further validation. This would further validate and/or refine the emulator. Given the reviewer's concerns, we suggest that such an experiment would be worthwhile.

For the second part of this comment, the method of history matching hinted at in the conclusion provides a framework to address the highlighted issue. This is an iterative approach that progressively removes regions of the parameter space in which the emulator

provides a poor match to observed data. In the remaining regions, the simulator is sampled again and new emulators are built within each region. This is a conservative approach that only rejects implausible regions. We are in the process of carrying out a full application of this method with NAME, but including it in this paper would make it unreasonably long. If the reviewer finds it appropriate, we could add a section in the Appendix explaining the methodology in more detail without performing it.

It is, however, important to be clear what the scope of this paper is. Our analysis is focused only on understanding the behaviour of the NAME model itself, and the influences of its parameters, not its relationship with reality. The emulators we have built can be used to investigate this relationship, using the method of history matching hinted at, and we are performing this investigation, but including it in this paper would not be feasible. We will attempt to make this distinction clearer.

Specific comments:
1) Use references with descriptions of how NAME works to shorten section 2.

Section 2 contains 3 paragraphs and provides the reader with information about NAME and the case study being considered. The authors believe that all this text is necessary.

 2) page 5, line 24: particle density and particle size distribution are not
known at the time of an eruption, so a forecast model must rely on a priori information.

We agree. Here we are highlighting the need for information about the volcano. Initially these ESPs not known or poorly known and a priori assumptions must be used until observational data is known. The text in Section 3 clarifies this.

3) page 13, line 17: only fine particles are considered, so why have particle size distribution as a variable, increasing the dimensionality, in this model (described in a later section).

In this context "fine" is not a constant value but a distribution between 0.1 and 100µm. While only the finer particles are considered, these are not all so fine that the size has no effect on the predictions. They need to be fine enough to reach the distal plume, but may still sediment significantly over the duration of the simulation. Here the particle size distribution is described using two parameters, shape and scale of a fitted gamma distribution which is a reduction from the use of 6 particle size bins.

4) page 14, line 20: 10,000 per hour, 1000 per hour are the rates of particle release, not the number of particles.

Agreed. The text on page 14, line 20 will be updated to reflect this.

5) page 16, lines 15-20: these are properties. So remove the word 'must' and replace 'should predict' with 'requires'.

Text will be updated to the following:
- It evaluates quickly
- It is expressive ...
- It predicts that ... are very close when ...

6) page 16: write out the function g, here equal to 'x' to the ith power.

This is incorrect. i indexes the basis functions and, at this point, these functions could be anything. Later on (section 6.1) they are chosen to be the functions $x_i$, $x_i x_j$ ($i \mathrel{/=} j$) and $x_i^2$ with i,j ranging over the inputs. No powers higher than 2 are involved - there are a lot of functions because there are a lot of components to the input vector x, not sa lot of different powers. Near end of p16 we will change the text to "Here, the $g_i(x)$ are chosen to be simple functions ..." to emphasise that the $g_i$ are a choice.

7) page 16: define both expectation and correlation rather than just putting in E() and Corr() and expecting the reader to just know what this means.

Agreed. This will be clarified in the text.

8) page 17: same comment as above for Var() showing up suddenly in the text.

Agreed. This will be clarified in the text.

9) page 17: line 25: 'some of these problems are summarized'. Nice lead in, except that none of these problems are summarized later on. I suspect they got abandoned in an earlier draft.

The problems we had in mind are discussed on p18, lines 16-24.

10) 'expert judgements' throughout are the a priori in your Bayesian analysis.
We agree (although the expert judgements can be hard to express in the form of probability distributions). We hope our agreement is clear in the passage from the last paragraph of 5.1 to the end of page 18. However we will amend page 18, lines 27-28 to try and make this clearer, referring to both "experts" and "prior" quantities in both sentences.

11) page 18: The paragraph/sentence starting with "Such an approach" is a non-sequitor with the paragraph and sentence immediately above. What approach?

We mean the approach that the previous paragraph describes - i.e. a full Bayes calculation. We are unsure why this isn't clear, but we could be more explicit with "A full Bayesian calculation of the type just described has been successful in many applications."

12) page 18, 1st paragraph: forecasting uses the posterior, which has the prior in it. Change 'For calibration and forecasting' to 'For calibration'

This is true in many Bayesian calculations, but things are more complicated with emulators. Bayes is being used to determine an approximation (i.e. the emulator) to the function f(x) representing the dependence of NAME's output on certain parameters. Here the NAME simulations are the "observations" which lead to the a posteriori model. At that point we have an approximation to NAME, but values of the parameters still need choosing if we are to make a prediction for concentration levels. This could involve a separate Bayesian calculation using e.g. satellite observations of the ash cloud. This is what the text very briefly refers to. In fact the paper only uses the emulator to better understand NAME's sensitivities and so the text here (last sentence of the paragraph) could be removed. However we think it helps give some wider context to what is being presented.

13) page18: The reason that doing high dimension problems with a linear Bayesian analysis is very difficult is that linear regression is not well suited to high dimension problems. You don't have enough flexibility with fixed basis functions. This could have been alleviated in this paper by just considering two parameters for distal clouds; height and mass flux at that height.

We mostly agree, but considering just two parameters would be a different paper with a different aim. Our aim here is to explore the uncertainties and sensitivities of NAME arising from a wide range of uncertain parameters. It is certainly possibly that we might not have had enough flexibility in the basis functions (which would put more importance on the residual functions u and u'), but this can be tested after the event by checking the emulator output against some NAME simulations which have not been used in the emulator.

14) page 20, first paragraph: so, a prior was not used here?

That's correct. To avoid any doubt we will change "standard (non-Bayesian) linear regression" to "... linear regression (without a prior)".

15) page 22: give references for R squared.

This is a very standard statistic as such it is not normally given a reference.  Text will be updated to "The coefficient of determination, $R^2$, which represents ...".

16) page 24, line 24: change "..., whereas of course" to "..., even though".

This change will be made in the text.

17) page 24, bottom: so, the number of parameters could not be reduced below 4? What if you did not do this? Would you just end up with height and mass flux? I suspect so.

If we removed the requirement to have at least four parameters in each model, the results would be as follows. In many of the regions the process would still select four parameters because the regressions were poor with fewer than four parameters. In the remaining regions, most would be reduced to 2 parameters and indeed these would be height and mass. In a couple of instances, a different parameter to mass would be chosen (height would be in every model), or three parameters would be chosen. However, most of the 2- and 3-parameter emulators would not pass validation.

18) page 34, Cov equations at top: there are 2 parameter sets here. Need either references or a proof of the second equation.

This is a consequence of $Cov(\Sum_i X_i, \Sum_j Y_j) = \Sum_{i,j} Cov(X_i, Y_j)$. However we agree that the text needs to be modified to make it clearer to the reader that u' is not random here and that we are considering only $x_j$ in $\chi_F$ (and so u' is known).

**Review 2**

The manuscript presents a complex and potentially very interesting application of emulation modelling to quantify the relative impact of several uncertain parameters on the predictions of a volcanic ash transport and dispersion model (the NAME model). The

main novelty of this study is that it presents one of the first applications of a formal sensitivity analysis (i.e. beyond one-at-the-time approaches) to a volcanic ash transport and dispersion model. Moreover, some of the techniques and lessons learnt in this study (for example on how to link fast and slow simulators) might be of interest to a broader community who deals with uncertainty in computationally expensive transport and dispersion models, not only volcanic ashes.

However, the manuscript in its present form is quite unclear and the structure unbalanced, which makes it difficult to fully evaluate and appreciate the findings of the study.

We thank the reviewer for taking time to review our manuscript and give such constructive feedback. We agree that the paper could be reorganised to make it clearer to the reader the key messages of the study.

Below my main issues and some suggestions for revision.

[1] The choice of contents in the methodology sections is confusing. Too much space is given to topics that are generic and covered in many other papers or even textbooks, which divert attention from and leave too little space to specific information about the NAME application. For example, on P. 14 L. 16-29 almost the same space is devoted to describing the difference between fast and slow simulators, which is a very important setting of the model under study (see also comment [3] below), as to describing the well-established and totally generic Latin Hypercube sampling technique.

We agree and the long section outlining the Latin Hypercube sampling technique will be reduced.

The description of emulators and Bayes linear methods covers 4 pages (P. 16-19) although much of the content is very generic, easily accessible elsewhere, and - most importantly -
not strictly related to the application here. In fact, from P. 20 L. 10-12 ("a successful method has been to use a standard (non-Bayesian) least-squares regression") I understand that the non-linear/linear Bayesian approaches discussed on previous pages are not actually used here because linear least-squares are sufficient.

The Bayes linear calculations are used later (see the next comment), therefore the Bayes linear section needs to be included but it will be revised to highlight the points needed to follow the narrative of the paper.

 As a reader I was confused by these long descriptions and diverted from focusing on the specific features  of this application, for example the link between emulators of the fast and slow simulators. Another example is Section 5.4: this is all generic and standard methods that do not need to be discussed in an application-oriented paper, especially if not used then in the Results section (see for example the comment on "examining the residuals"...).
In summary, I would recommend to deeply revise these sections, to make them shorter and include only the information relevant to the specific application and thus needed to understand the results.

We agree that the balance should be improved. As noted by the reviewer, the two main focuses of the paper are, or should be, the application to NAME and the use of multiscale emulation. The other parts of the methodology sections can be significantly shortened to

improve clarity. To further address one point in this comment, "I understand that the non-linear/linear Bayesian approaches discussed on previous pages are not actually used here because linear least-squares are sufficient", while it is true that we use the least-squares fit to fix most of the parameters of the emulator for the fast simulator, there is still a Bayes Linear adjustment performed to adjust the emulator for the slow simulator. Therefore, some discussion of Bayes linear is necessary, but can be reduced (and we can specifically clarify exactly where the Bayes linear adjustments are happening and where they are not).

However, since the first reviewer would like some expanded detail in some of these areas, it could be a better idea to retain and possibly expand some of this material, but to move it to the Appendix.

[2] The results section seems to focus a lot on the validation of the emulators - i.e. how good they are in representing the fast and slow simulators - and their similarity, rather than the ultimate goal of the analysis, which is to use the emulators "as a research tool to better understand the simulator, the role of the parameters, the interactions between them..." (P. 4 L. 17). For example, while Fig. 5 and 6 report validation results for the emulators, there are no Figures reporting sensitivity results, parameter mapping or interactions. The only results related to sensitivity analysis are in Table 4, which however does not even use the word "sensitivity"! This is odd especially considering the title of the manuscript.

We agree that there is a large focus on the validation of the emulators. However, it is important to bear in mind that the emulation procedure is based on a lot of choices (basis functions, correlation function, correlation distance, and so on) and that it is the validation step that gives confidence that the choice made were good ones. Therefore we need to make sure we include enough information that readers can have confidence that the emulators actually work. To reduce the amount of space dedicated to this in the paper Fig 6 will be removed and some details of the validation process will be placed in the appendix.

We will add specific details of the expectations and variances of the adjusted coefficients in the model, the ranges of predictions that can be generated from the parameter ranges, and some plots of the emulator predictions

[3] The difference between fast and slow simulators should be better clarified. In particular, on P. 14, L. 16-24: What does the "increase in particle-sampling noise" mean in simpler terms? How does it impact the simulation results? Not so much - at least for predicting average column loadings - according to what the authors say later on P. 15 (L. 7-12).

Particle sampling noise is due to the stochastic motion of the particles within NAME and the consequent randomness in the number of particles in a given region, and hence in the total particle mass in a given region. The fractional noise is proportional to 1/sqrt(N) where N is the number of particles. Here the impact is small as we average over large regions. Text will be added to the manuscript to clarify this.

On the other hand, how fast is the fast simulator?
These aspects need to be clarified because they are at the basis of the motivation of the study: if the fast simulator is fast enough, and its predictions of the output of interest (average column loading) are reasonably close to those of the original (slow) simulator, then why building an emulator? One could directly apply a Global Sensitivity Analysis technique (for instance, the Morris method, or Regional Sensitivity Analysis, which are

both reasonably "low-cost") to the fast simulator. I am not saying that this is necessarily the case (maybe the fast simulator is not so fast, or there is some other reason I am missing to avoid using it) but more details should be provided to clarify this crucial point.

The fast simulator takes between 10 and 20 minutes to run. So much faster than the "slow" simulator but not quick enough to apply the techniques you suggest. Text will be added to clarify this.

[4] Choices underpinning the analysis and their impact on sensitivity results needs further discussion. P. 6, L. 20 onwards: there are three set of choices that are made here: - the choice of the uncertain parameters to be included in the sensitivity analysis (out of a larger set of parameters appearing in NAME, which are set to default values - two of them are further described in Sec. 3.2.7 and 3.2.8, together with the reasons for not varying them - what about the others?) - the choice of plausible ranges for the uncertain parameters - the choice of a particular event, and hence a particular set of forcing data, for the model simulation (and the choice of ignoring the uncertainty in those data) I presume that these choices may have a very strong impact on the results and thus on the generality/transferability of the findings. Assessing such impact might be beyond the scope of this manuscript, but the point should be at least mentioned and discussed.

In this study the input meteorology was not varied because the large dimensionality of the met input makes it unsuitable for this type of approach. Also at the time an appropriate set of ensemble meteorology was not available. We accept that this could have a large impact on the ash column loadings, although in the case study presented the meteorological situation is relatively settled. However this does not reduce the value of understanding the uncertainty due to other causes, while recognising that this is not the total uncertainty. Text will be added to the conclusions to clarify this beyond the scope of the paper. Expert elicitation identified the parameters leading to uncertainty and where a parameter is set to the default value it is highlighted.

The parameters chosen to be included in the sensitivity study, along with their plausible values, were determined through an expert elicitation exercise. Full details of this procedure are beyond the scope of this paper but they involved consideration of estimates for these parameters found in the literature, choices made by various experts who run this and similar models, and from the personal experience of the co-authors from the Met Office who have significant experience with NAME. A brief discussion of this can be added. If the reviewer considers this aspect particularly important, then we would be happy to provide an expanded discussion in the appendix. Section 3 presents these parameters and where a parameter is set to the default value this is highlighted. From the perspective of understanding the sensitivities, the exact ranges are not crucial. The ranges are more important if we try to infer uncertainty directly from the ranges, but the intention is not to do that but to infer more suitable ranges from history matching.

We accept that we have only presented one case study and that this has limitations although we feel we have not brushed over this. The conclusions already contain a sentence "These conclusions should be tested in other situations to assess how widely they hold" (Page 30 L16). The same paragraph also starts with "For this case".

OTHER SPECIFIC POINTS:
P. 3 L. 28: "Finally, the analysis cannot provide ...". A bit vague, please clarify what is an "overall assessment of uncertainty"?

The uncertainty from the use of a computer simulator comes in very many forms, for example parameter uncertainty, measurement uncertainty, uncertainty about the effects of assumptions within the model, uncertainty about the impact of missing processes, and so on. The primary uncertainty being considered in this paper is the uncertainty associated with the value of the simulator at a set of parameters at which it has not been run. The simple analysis being discussed in the introduction does not lend itself to a numerical representation of this uncertainty. In contrast, the emulation method contains exactly such a representation, which could then be combined with all other assessments of uncertainty to provide an overall assessment of uncertainty.

P. 8 L. 3-4: Difference between "full depth" and "thin layer" is not clear to me. Also, very unclear how the 1700 + 1700 runs here mentioned are connected with the 1500 + 200 runs mentioned on P. 14 (the same experiment of P. 14 is repeated twice, once per each source type?). Maybe the confusion could be avoided by simply not giving all these details on the thin layer source case, since its results are not shown (as commented on P. 23 L. 15-16)?

When initially designing the study two sets of runs were performed to test the impact of the vertical distribution of ash within the ash column. The majority of the references to this will be removed and replaced with a "the same analysis was performed with the thin layer source, with no significant differences in conclusions (although the emulators in this case performed slightly worse consistently)".

P. 10 L. 18: " are varied by the same proportion". Unclear.

We agree this is unclear. The text will be revised to clarify this.

P. 12 L. 23: "between 0 and 2": or between 0.5 and 2 (assuming this section refers to $x_{17}$ in Table 1).

The scaling factor does vary between 0.5 and 2. The text will be updated to reflect this.

P. 14, L. 10-11: "the average ash column loading predicted ..." not completely clear. Are column loadings per each region averaged over the simulation period, thus defining 75 "outputs" (and 75 emulators), or are predictions for each hour analysed separately (thus defining 75xT "outputs", where T is the number of hours in the simulation)? Please clarify, maybe also inserting an equation here.

There are 75 emulators in total. There are up to four regions each hour for the 24 hours simulated. The regions are chosen using the satellite retrieved ash column loadings so the regions change from hour to hour. All regions are shown in Table 3. The text will be revised to make this clearer. Note that the NAME column loadings are averaged over the hour (not snapshots).

P. 14, L. 14: Again unclear: "regions used for the first hour are marked..." So, the definition of regions changes from one hour to another? And also their number then? And so how does it connect to my previous question?

Please see comment above.

P. 15, L. 7-12 and Figure 3. Again on the difference between fast and slow simulators. I understand that the main conclusion here is that simulations from fast and slow simulators are similar, however this paragraph and the Figure are rather unclear. "agreement between the two simulators" means that they provide similar predictions of average column loading? What does "to be related but not in agreement" mean? The "correlations" of 0.99 and 0.7 are the correlation between simulated column loadings (in different regions? at different time in the simulation period?). Please be more specific. Figure 3: units of measurements on the axes are missing!!! (I guess they are "Log ash" as in Fig. 5?).

The correlation range is the range over all 75 regions that are emulated over the whole simulation. Each correlation refers to a single region and time, and that the correlation is over the 200 different choices for the parameters. There are 200 points in Fig 3a and 3b. The text will be revised to make this clearer and the "related but not in agreement" statement will be rephrased to "correlated but not in close agreement".

The intention was to say that for some of the regions there is an almost exact match between fast and slow simulators, whereas in others there are noticeable differences between the predictions of the two models but there is still a clear relationship (this can be of two forms: in some regions, if one parameter set has higher output than another in the fast simulator, it will have higher output in the slow simulator, but the values will be different; in other regions, this is usually the case but sometimes there will be lower output (e.g. the second plot in Fig 3). The highlighted sentence is awkward and must be changed. We will also fix the units in Fig 3 (and improve the axis labels in general in most plots)!

P. 15, L. 7: "the goal" ... This is a bit misleading: the ultimate reason for building the emulator is not making inference, but rather understanding the role of parameters (see discussion on P. 4 L. 14-17). Maybe good to remind it here.

Yes, we agree this is a bit misleading. This is referring to the goal of the statistical technique not the overall study. The text will be revised to clarify this.

P. 28, L. 9-10: "for some parameters... it is not plausible": so you included in the analysis some parameter combinations that are implausible? This sounds contradictory (who would be interested in sensitivity to parameter variations that are not plausible?). Please clarify

We agree that it is not sensible to investigate parameter sets that are not plausible. This sentence was included to reiterate that the maximum turbulence values used in this study are representing plausible extreme values of turbulence (P10 L16-17). It is not expected that these values would be present everywhere in the atmosphere but the current model parameterisation uses a constant value of turbulence in the free troposphere. To model spatially/temporally varying a new parameterisation would need to be developed. Text will be added to clarify this point.

P. 29, L. 18-23: This is a generic comment that was already made in the Introduction (P. 3) and is not part of the results. I would remove it.

We agree. This section of text will be removed.

Table 1: - term "default" on first row is a bit misleading - does it refer to the "data from Keflavik radar" (as explained on P. 7, L. 19)? If so, clarify in the Table - term "default" on third row is a bit misleading - does it refer to the MER value from Eq. (1) (as explained on P. 8, L. 20)? If so, clarify in the Table - connect better to the text, for example the mathematical symbols used in the text could be included in the

description of "Parameter name" (for example R_a on row 17...) - rows 7-10: "varied in proportion..." is unclear (here and in the text) - row 18: so the default value coincides with the maximum value? This looks strange.

Default in row 1 will be replaced with "Arason et al.".
Default in row 3 will be replaced with "Mastin et al. "
As suggested the mathematical symbols for the parameters will be included in the "Parameter name" column.
For clarity text "varied in proportion with" will be removed from the body of the table but text will be added to the caption to highlight the linkages. .
The default value being set as the maximum value is what was decided upon during the expert elicitation exercise.

Figure 3: Caption mention "(a)" and "(b)" but letters are not reported in the panels.

This inconsistency will be corrected in the revised manuscript.

---

## Author Response (AR2)

**Editor Decision:** Publish subject to minor revisions (further review by Editor) (05 Jul 2017) by Thorsten Wagener

We thank the reviewers and Editor for taking the time to review our manuscript again. Your comments have definitely improved the clarity of the paper. Editor/reviewer comments are black. Responses to comments by the authors are blue. Note that line references refer to the most recent version of the paper.

Comments to the Author:

The reviewers have gone through the revised manuscript again, and, while they accept that things have improved, they still have quite a few comments about how to make the manuscript more accessible and clearer. I selected minor revisions for now. However, I still expect you to address all the points that I list for bother reviewers in detail. All these comments should make the manuscript easier to understand and therefore will help you. So please take these comments seriously. I will send the newly revised manuscript out to reviewers again if I do not feel that you have done so. There are 7 comments by reviewer 1 and 3 main comments by reviewer 2, with some additional minor ones.

We think reviewer #1's comments 2, 3, 4, 6 and 7 reflect a misunderstanding of the aim of the paper. We've tried to explain this in more detail below and have modified the paper to try to prevent such misunderstanding.

**Report #1**

This paper should be revised significantly before publication. Some information is missing, such as an equation showing how model results are differenced with observed data. There are typos and some phrasing indicating hurried preparation. My comments are as follows:

Scientific comments:

1)      Line 5, pg 9; use of equation 1 relates x1 to x3. Constraining one parameter constrains the other. With a line source (line 18, pg 8) this is presumably distributed uniformly from surface to top of plume.

Yes, your interpretation is correct.  This is noted on P8, L25 and P9, L5. We have changed "evenly" to "uniformly"  for clarity.

2)      There are 18 parameters to constrain, and only the distribution of cloud load with time to constrain them. Perhaps the scheme acts to just confine the fit to the data manifold, and so impotent parameters are ignored. But the number of parameters does seem superfluous. For example, do the authors need two turbulence parameters and two loss parameters, when they only have scalar transport in a given wind field?

The aim of this study is to better understand the influence of source and internal model and parameters on the simulator (NAME) output (horizontal distribution of ash column loading).  To do this we need to include simulator parameters, in this case this includes two turbulence parameters and two loss parameters. Note that similar emulation techniques have been successfully applied to a variety of physical problems in the literature.

We emphasise that our aim is not to *constrain* parameters (e.g. using actual atmospheric observations of the ash cloud), but to build an emulator to replicate the model behaviour. With enough runs of the model there is no problem in principle of understanding the influence of any number of parameters on column load. We've tried to make clear the difference between constraining parameters with real observations and building an emulator to approximate the model (see p4 line 27 to p5 line 3, p5 lines 21-25, and p17 lines 3-4, p19 line 8, p22 line 24 and p24 line 5).

3)      Lines 10-17, pg 13; it seems that including the laminar sublayer (very near the earths surface) is a very different scale than the transport of ash in the atmosphere at an altitude of kilometers above the surface. Why include this? By the time ash is near the ground, transport is local and on a scale for which the authors have no data.

We are aiming to understand the influence of as many of the internal simulator parameters as possible so have included this term for completeness. We don't understand the comment about the authors having no data on a local scale. We have no actual atmospheric ash cloud data at all in this study – the aim is to build an emulator to replicate the simulator.

4)      Equation 8; this is a linear regression equation with an error u(x), which is then used later in a linear Bayesian analysis. This is fine, but the means by which data are compared to model runs to perform this fit is not mentioned. Do the authors use a quadratic loss function, for example?

In section 5.1 we specify that least-squares regression is used to construct the emulator model in Equation 8 for the fast simulator. The emulator for the slow simulator is fit using a Bayes linear update described in Section 5.3.

We note we are not comparing real atmospheric data with results from (NAME) simulator runs, but are comparing data from runs of the NAME simulator with data produced by the emulator (and, for the fast simulator, are choosing the beta to minimise the sum of the squares of the differences).

5)      Pg 21; Again, the authors are using a linear regression, with R squared, but do not explain what differences are used. The form of the equations in the Appendices suggest a quadratic loss function, but the reader should not have to guess.

It has been restated in Section 5.4 and at the start of section 6.2 (and at many other places in the text) that least squares is being used.

6)      Lines 10-18, pg 25; the curse of dimensionality needs to be discussed somewhere here. How can so few data (distribution of cloud load with time with a given wind field) be constraining interactions between so many parameters?

Perhaps this comment and comment 6) below under Editorial comments suggest a misunderstanding of the emulator's purpose: it is trying to predict what the NAME simulator will do under given conditions and parameters. It does not, by itself, provide any predictions about real-world behaviour. Emulators are designed to understand what the simulator would do if we ran it at new parameter settings.  Given enough runs of the model there is no shortage of data for understanding how the model's column load varies in response to any number of parameters. We have added the following into the introduction: "It is important to understand that emulators are used to model the behaviour of the simulator itself, when parameters are varied. That is, an emulator is designed to predict the output of the simulator under given conditions. The relationship between the simulator output and real-world observations does not have to be considered in order

to build an emulator; the ``observations'' used to build the emulator are observations of simulator output, not real-world measurements."

7)      The two most important parameters, plume height and source strength, are tied together by equation 1 and do not vary independently. With any significant wind shear with altitude, there is only a limited range of solutions to be obtained as the line source increases in height, and much of the ash delivered at various heights are spurious sources. The heights are tied together by the nature of the chosen source, and these combined heights spew ash in different directions as the wind varies with height. The different directions are tied together, producing a single solution to be compared with data. With this source, the emulator cannot separate out the effects of different heights, but will use the degrees of freedom provided by the different parameters to try and explain the variations in distribution. A more effective approach would be to consider point sources at specific heights, and allow their strength to vary independently of height. Ash is usually delivered at a narrow range of elevations, despite the large column heights at the source.

We thank you for your comment and suggestion of approach. As stated on P8, L18-22, we have performed emulation procedure using a "thin layer" source (where all the ash is released close to one height) and in this case there is very little difference in the final results on parameter sensitivities.  We emphasise that we are not trying to compare with observational data or to explain observed variations in distribution of ash in the real atmosphere. However, if we were doing this, we agree that the height distribution at the source deserves more consideration.

Editorial comments:

1) Line 15, pg 7; 14 parameters mentioned here, 18 in Table 1

This inconsistency has been corrected in the text.

2) Line 15, pg 10; turbulent

This has been corrected.

3) Line 12, pg 12; B is used as a parameter here, and B is used for something different on pg 19.

For clarity and to emphasise the difference between the variables we have changed the font of the B used on P12.

4) Line 20, pg 13; D is used as particle diameter here, and D is used as a vector of simulator runs on pg 19.

For clarity and to emphasise the difference between the variables we have changed the font of the D used on P13.

5) Line 14, pg 17; hosen

This has been corrected.

6) Pg 16; this might be a good place to put exactly how the comparison between model cloud load and observed cloud load is used to construct a vector of differences; is it quadratic? Absolute value? How are simulator outputs compared with data?

See response to scientific comments 4 and 6 above.

**Report #2**

Suggestions for revision or reasons for rejection (will be published if the paper is accepted for final publication)

The manuscript has been significantly improved after revision. I think the contribution is interesting and should be considered for publication. However, the manuscript could still be improved in terms of clarity. Below are details of 3 major and several minor suggestions for improvement.

[1]     Clarify the connection between emulator development and sensitivity testing of the slow/fast simulators. One may expect the sensitivity testing to be carried out in two stages: first identifying the emulator, then using the emulator as a substitute of the simulator within a computationallyexpensive sensitivity analysis. However after reading the entire manuscript I understand that there is no such a second step, and that the sensitivity testing is a "byproduct" of the emulator identification process itself: the estimated emulator's coefficients (beta) are the measures of the simulator sensitivity to its parameters (x). This could be clarified in the Abstract and Introduction. For example on P. 4 L. 18-19: "This enables the quantification of the impact of each simulator parameter on the prediction of the dispersion of volcanic ash": the sentence could be revised to clarify "how" the quantification is enabled.

On P4 we added "In this study, the impact of the various simulator parameters can be assessed by their coefficients within the emulator. Since the emulator can be evaluated quickly, it can also be used to replace the simulator in any computationally-intensive sensitivity analysis method of choice, though this step is not performed in this study."

On the same topic, the term "active" should be defined in Sec. 6.1 (how is it operationally concluded that a parameter is "active"?) and the link to sensitivity testing established. I understand that the fact that a parameter is active in the emulator implies that the output of the original simulator is sensitive to that parameter, so "finding active parameters" is the sensitivity testing, but this is not stated explicitly. Please clarify.

We have included a definition of active and inactive variables at the start of Section 6.1 and a link to the section of the appendix where the specific details of the process of identification is explained. We also clarified that active variables are not necessarily really impactful: "Note that active variables are not necessarily extremely important parameters; they simply provide some information that would be lost by excluding them."

Identifying inactive/active parameters could be regarded as part of the sensitivity testing, but it's only part, and it leads to a smaller set of parameters for more quantitative analysis.  Alternatively one could view it simply as a step in building the emulator. The sensitivity analysis can then be done using the emulator, either just using its beta values as is done here (these can be regarded as zero for inactive parameters, making the sensitivity analysis for these parameters trivial) or by using the full emulator. Our presentation tends to emphasise this second view which we think is clearer.

[2]     The description of the emulation methodology (Sec. 5 and associated Appendices) is still confusing. I think the main text still puts relatively too much emphasis on the statistical rationale underlying the emulator and too little on the practical steps for their construction and use. This may

make the manuscript not very accessible to non-statisticians and limit the uptake of the proposed multi-level emulation approach.

We have added in the appendix a step-by-step description of the process used and, for each step, references the section(s) of the paper dealing with the step.

Specifically:

-        P. 18 L. 9: "Such an emulator then provides predictions for f (x) at a new x. Since it is a statistical model, this prediction also comes with an associated uncertainty". How are the predictions and associated uncertainty obtained in practice? I guess the prediction is the adjusted expected value of Eq. (9) and the associated uncertainty is expressed by an uncertainty interval based on the adjusted variance of Eq. (10), is this correct?

In the Bayes linear framework used in this paper, the prediction takes the form of the expected value of f given the results of the collection of simulator runs, and the associated uncertainty is the variance of f given the simulator runs

 Also, what is the link between Eq. (9)-(10) and Eq. (8)? Is the expected value E(B) in Eq. (9) given by the first term in Eq. (8) (the linear combination of "g" functions)? What about the variance? Please clarify.

We have added a specific example in the Bayes linear section of what E{B} and so on mean for the fast emulator. In the most general case, E(B) would be (sum E(beta)_i g_i(x) + E(u(x))), and Var(B) would be Var(sum E(beta)_i g_i(x) + E(u(x))) and hence include lots of correlations terms between the betas and u(x) which would have the be specified a priori. In our application, much of this is skipped by taken E(beta_i) as the regression estimates, Var(beta_i) as zero, E(u(x)) as zero, and Var(u(x)) as the residual variance, and leaving only Corr(u(x_1), u(x_2)) to be modelled (and even this is done using the data rather than specifying a prior).

-        P. 19 L. 1 says that the Bayes linear approach is used for "the analysis of the link between the fast and slow emulators". Does this mean that Eq. (9)-(10) are also used to establish a link between the two emulators? If so, does this mean that the prediction of one emulator is adjusted based on the prediction of the other? This seems in contrast with Sec. 5.3, from which I understand that the link between the two emulators is established by linking their respective coefficients beta. Again this should be clarified.

Yes, there is a second Bayes linear adjustment used in the section. The link between the emulators is modelled as a link between the betas, and the Bayes linear adjustment is used to update this link based on observations. In the first paragraph of Section 5.3 this further use of Bayes linear has now been explicitly stated. The second paragraph in this section now also makes explicit that Bayes linear is being used to learn about the u'(x), the rho_i, and also the w(x) (that is, the local variation in the fast emulator, the link between fast and slow emulator coefficients, and the remaining local variation in the slow emulator).

-        P.18 L. 4: "Conceptually, the expectation, variance, and correlation are a priori uncertainty judgements". What does this exactly mean? Which of the 3 parameters (expected value, variance and correlation length) are actually estimated from the residuals of the emulator predictions and which are assumed by a priori judgement?

Given the restructuring in the last revision, this sentence is not really meaningful anymore, so has been removed.

We've added some text (p17 line 10) to provide a general framework and make clear that the emulator structure may be designed using a mixture of judgement, exploration of the data and tuning. We've also removed the sentence referred to by the reviewer about the three specific parameters. The precise way the three parameters are chosen for our specific problem can't really be explained until some more ideas (Bayes-linear, fast and slow emulators) have been introduced, but we hope the added text will help the reader understand the range of possibilities.

-        How is the "correlation length" estimated? The description in A2.2. is unclear: what does "tune" mean on line 25 P. 34? Manual tuning? How is it checked that the method "has been successful" L. 1 P.35?

We have expanded this section to explain the method more clearly. A subset of the observed data is used to find a correlation length that provides accurate estimates with as low variance as possible. The full data set is then used to check that the correlation length gives good predictions on the entire data set.

 [3] From Figure 3 it seems that the difference between the fast and slow simulators is really small (at least for the chosen output variable). In the best case (top panel) the two simulators produce almost identical output, in the worst case (panel (b)) the outputs are still well related to each other (only few points in the bottom left part of Fig. 3.b would not align to a simple interpolating line). I guess this is the reason why the two emulators are found to be very close (P. 24 L. 4-6: "the link between mean functions of the two emulators is strong and consistent") and their difference "mostly a rescaling". I think the similarity of the fast/slow simulators should be emphasized more when commenting the emulator results, also to acknowledge that this case study application may not be the most challenging one to test the multi-level emulation approach (although the idea remains valid and very interesting in principle).

On P24 we have added "Given the similarity between the two simulators, it is not surprising that the multi-level emulation method works smoothly in this application; in applications with more fundamental structural differences between the simulators, it is likely that more careful modelling of the link would be required." To draw attention to this point.

MINOR

P. 4: maybe remove the line break between line 2 and 3 - the sentence "Finally, ..." should be part of the previous paragraph.

This has been corrected.

P. 4 L. 7-8: "The emulation method that is presented in this paper gives assessments of uncertainty that can be combined easily with other sources." I agree this is possible in principle but not actually demonstrated in this paper. Maybe good clarify the point.

This is a good point, we have added "(although actually performing such an assessment and combination is beyond the scope of this paper)" to this sentence.

P. 4 L. 10: "is expensive in both time and money." A bit vague. If I understand correctly, the point here is that sensitivity tests of a wide range of parameters require a lot of model runs and for a complex simulator such as VATD this would take a lot of computing time.

We agree this is vague and have added "to perform such an analysis requires very many simulator evaluations and hence very large computation time" to this sentence

P. 5 L. 15: "the emulators used..." remove "used"?

This has been corrected.

P. 5 L. 21-22: "Section 5 gives an overview of the statistical methods used in the analysis." Maybe specify this is about building and evaluating the emulators

We agree. This has been changed to "gives an overview of the statistical methods used to build and test the emulators."

P. 10 L. 2: "all the alternative PSDs could be reconstructed to a reasonable approximation": unclear. What is "reasonable approximation"? Does it mean that the alternative PSDs are compatible with the range of observations by Dacre et al. (2013)?

We agree that this is unclear. The observed PSDs presented in Dacre et al. (2013) can all be reconstructed by varying the shape and scale parameters in a gamma distribution. The text has been updated to make this more clear.

P. 17 L. 14: "hosen" should be "chosen"?

This has been corrected.

P. 17 L. 15: "For the rest of this section, attention is restricted to scalar-valued for simplicity of notation." Ok but please clarify whether in your application a vector-valued emulator was used.

We agree this could be clearer and have added "For the application to NAME, f is vector-valued but this is handled by constructing separate scalar emulators for each f_i."

P. 18 L. 20: "It" should not be capital letter

This has been corrected.